# Recurrent Action Transformer with Memory

## Abstract

Transformers have become increasingly popular in offline reinforcement learning (RL) due to their ability to treat agent trajectories as sequences, reframing policy learning as a sequence modeling task. However, in partially observable environments (POMDPs), effective decision-making depends on retaining information about past events – something that standard transformers struggle with due to the quadratic complexity of self-attention, which limits their context length. One solution to this problem is to extend transformers with memory mechanisms. We propose the **Recurrent Action Transformer with Memory (RATE)**, a novel transformer-based architecture for offline RL that incorporates a recurrent memory mechanism designed to regulate information retention. We evaluate RATE across a diverse set of environments: memory-intensive tasks (ViZDoom-Two-Colors, T-Maze, Memory Maze, Minigrid-Memory, and POPGym), as well as standard Atari and MuJoCo benchmarks. Our comprehensive experiments demonstrate that RATE significantly improves performance in memory-dependent settings while remaining competitive on standard tasks across a broad range of baselines. These findings underscore the pivotal role of integrated memory mechanisms in offline RL and establish RATE as a unified, high-capacity architecture for effective decision-making over extended horizons.

## 1 Introduction

Originally developed for Natural Language Processing (NLP), transformers [50] have recently demonstrated strong performance across a wide range of Reinforcement Learning (RL) settings [1, 30]. They have been successfully applied to online [36, 15], offline [10, 23, 52], model-based [9, 42], and in-context RL [40, 17, 43]. In particular, transformers show promise for tackling long-horizon credit assignment and operating in memory-intensive environments [34, 17, 15, 36], provided the full trajectory fits within the model context. Despite their success, transformers face fundamental limitations when applied to long sequences due to the quadratic complexity of self-attention [25], which restricts their applicability in long-horizon inference tasks. While various techniques have been proposed to extend the context window [13, 7], these approaches often suffer from training instability [54] or rely on task-specific sparse attention patterns that may not generalize well beyond NLP [6, 53]. Memory-augmented transformers offer a promising alternative by enabling access to past information without expanding the context length. Motivated by advances in memory mechanisms for NLP models [13, 7], we investigate how such approaches can be adapted to RL. Unlike NLP, RL involves structured and modality-rich inputs – observations, actions, and rewards – that require domain-specific encoding, and frequently exhibit high sparsity in both reward signal and observations.

In RL, memory usually refers either to using past information within an episode [27, 34], or to transferring experience across environments [24, 48], aiding generalization, sample efficiency, and Meta-RL [14, 51], and we focus on the former.

We introduce the **Recurrent Action Transformer with Memory** (**RATE**; see Figure 1), a memory-augmented transformer that incorporates three complementary mechanisms: learned memory embeddings, recurrent caching of past hidden states, and a novel **Memory Retention Valve** (**MRV**) for selective information flow. We empirically show that memory mechanisms effectively preserve information from previous steps, allowing the model to use past information when making decisions in the present. MRV is designed to control the process of updating memory embeddings and prevent the loss of important information when processing long sequences, thus enabling the processing of highly sparse tasks. To assess the effectiveness of our memory mechanisms, we conduct extensive experiments across a diverse set of memory-intensive environments, including ViZDoom-Two-Colors [46], Memory Maze [38], Minigrid-Memory [11], Passive T-Maze [34], and POP-

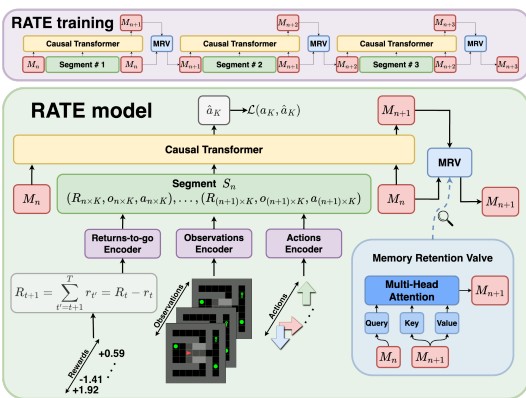

Figure 1: Recurrent Action Transformer with Memory (RATE). The model processes trajectory divided into $n$ segments $S_n$ with memory embeddings $M_n$, where $R$ denotes returns-to-go (future rewards), $o$ – observations, $a$ – actions, and $M_n$ – memory embeddings attached to each segment $S_n$ to retain important historical information.

Gym [32], as well as standard RL benchmarks such as Atari [5] and MuJoCo [16]. We also study the impact of memory on the performance of the proposed model. RATE interpolates and extrapolates well outside the transformer context and is able to retain important information for a long time when operating in highly sparse environments.

Our main contributions are as follows:

1. We propose **Recurrent Action Transformer with Memory** (RATE), a new transformer for offline RL that combines three complementary memory mechanisms: (i) memory embeddings, (ii) caching of hidden states, and (iii) a **Memory Retention Valve** (MRV), which uses cross-attention to retain key information over long horizons (see Section 3).

2. We conduct extensive evaluations on memory-intensive tasks – including ViZDoom Two-Colors, Memory Maze, Minigrid-Memory, POPGym, and Passive T-Maze – showing that RATE consistently outperforms strong baselines (see Subsection 4.1).

3. We further show that RATE matches or surpasses standard baselines on the Atari and MuJoCo benchmarks, demonstrating strong generalization across task types and highlighting the model's versatility (see Subsection 4.1).

## 2 Background

**Offline RL.** In RL [47], a task is formalized as a Markov Decision Process (MDP): $\langle \mathcal{S}, \mathcal{A}, \mathcal{P}, \mathcal{R} \rangle$, where $s \in \mathcal{S}$ are states, $a \in \mathcal{A}$ are actions, $\mathcal{P}(s'|s,a)$ is a transition function, and $r = \mathcal{R}(s,a)$ is a reward function. States satisfy the Markov property: $\mathbb{P}(s_{t+1}|s_t) = \mathbb{P}(s_{t+1}|s_1, \ldots, s_t)$. A trajectory $\tau$ of length $T$ is a sequence $(s_0, a_0, r_0, \ldots, s_{T-1}, a_{T-1}, r_{T-1})$, where $r_t = R(s_t, a_t)$ is the immediate reward at the timestep $t$. The return-to-go [10] $R_t = \sum_{t'=t}^{T-1} r_{t'}$ is the sum of future rewards from $t$. The goal is to learn a policy $\pi$ maximizing the expected return. While online RL iteratively collects trajectories through environment interaction, offline RL uses a fixed dataset of trajectories, making it suitable for scenarios where environment interaction is costly or risky. A popular offline RL method, Decision Transformer (DT) [10], models return-conditioned trajectories with a GPT-style architecture, avoiding value estimation. However, its fixed context window limits performance in tasks with delayed rewards or long-term dependencies, motivating memory-augmented models.

**POMDP.** In real-world, agents often receive partial observations rather than full states, breaking the Markov property. For instance, a robot using only camera input or an agent relying on past context. Such cases are modeled as Partially Observable MDPs (POMDPs): $\langle \mathcal{S}, \mathcal{A}, \mathcal{O}, \mathcal{P}, \mathcal{R}, \mathcal{Z} \rangle$, where $o \in \mathcal{O}$ are observations and $\mathcal{Z}^a_{s'o} = P(o_{t+1}|s_{t+1} = s', a_t = a)$ defines the observation function. Since single observations are insufficient, agents must use history to infer useful state representations.

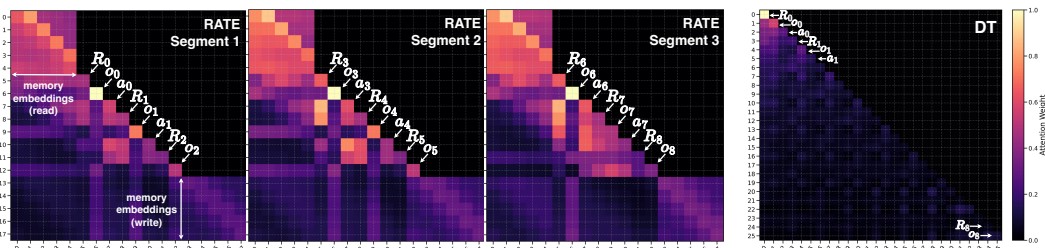

Figure 2: Attention visualization of RATE and DT on the T-Maze [34] task with a corridor length of $T = 8$. DT is trained on full 8-step trajectories, while RATE processes the sequence in three segments of length 3 recurrently, passing information between segments through memory embeddings.

## 3   Recurrent Action Transformer with Memory

Transformers excel at sequence modeling, including offline RL [10, 22], but struggle with long-horizon tasks due to fixed context and quadratic attention cost. In memory tasks, agents must recall information seen thousands of steps earlier—something models like DT cannot do once cues fall outside context. We propose the **Recurrent Action Transformer with Memory (RATE)**, which introduces segment-level recurrence and dynamic memory control. RATE processes trajectories in segments, using lightweight memory and a learnable **Memory Retention Valve (MRV)** to decide what to retain or discard. In T-Maze [34], the agent receives a one-bit cue $o_0$ at the first step indicating whether to turn left or right at the end of a maze. Solving the task requires remembering this cue despite sparse rewards. DT fails once $o_0$ leaves the context, making retrieval at inference impossible. Figure 2 shows this: DT attends to $o_0$ only when it fits the context, while RATE segments the input and propagates the memory embeddings, preserving the cue to the end and enabling explicit memory retention.

RATE combines memory embeddings [7], cached hidden states [13], and a novel MRV to handle long and sparse sequences. The architecture is shown in Figure 1. Let a trajectory $\tau_{0:T-1}$ of length $T$ be represented by triplets $(R_t, o_t, a_t)$, where $R_t$ is the return-to-go, $o_t$ the observation, and $a_t$ the

---

**Algorithm 1** RATE

**Require:** $R \in \mathbb{R}^T, o \in \mathbb{R}^{d_o \times T}, a \in \mathbb{R}^T$
1: $\tilde{R} \leftarrow \texttt{Encoder}_R(R)$
   $\tilde{o} \leftarrow \texttt{Encoder}_o(o)$
   $\tilde{a} \leftarrow \texttt{Encoder}_a(a)$
2: $\tau_{0:T-1} \leftarrow \{(\tilde{R}_t, \tilde{o}_t, \tilde{a}_t)\}_{t=0}^{T-1}$
3: $M_n \leftarrow M_0 \sim \mathcal{N}(0,1)$
4: **for** $n$ in $[0, T//K - 1]$ **do**
5:   $S_n \leftarrow \tau_{nK:(n+1)K}$
6:   $\tilde{S}_n \leftarrow \texttt{concat}(M_n, S_n, M_n)$
7:   $\hat{a}_n, M_{n+1} \leftarrow \texttt{Transformer}(\tilde{S}_n)$
8:   $M_{n+1} \leftarrow \texttt{MRV}(M_n, M_{n+1})$
     **Output:** $\hat{a}_n \rightarrow \mathcal{L}(a_n, \hat{a}_n), M_{n+1}$
9: **end for**

---

**Algorithm 2** Memory Retention Valve

**Require:** $M_n, M_{n+1} \in \mathbb{R}^{m \times d}$
1: $\mathbf{Q}_h \leftarrow M_n \mathbf{W}_Q^{h\ T}$
2: $\mathbf{K}_h \leftarrow M_{n+1} \mathbf{W}_K^{h\ T}$
3: $\mathbf{V}_h \leftarrow M_{n+1} \mathbf{W}_V^{h\ T}$
4: $M_{n+1}^h \leftarrow \texttt{softmax}\left(\frac{\mathbf{Q}_h \mathbf{K}_h^T}{\sqrt{d}}\right) \mathbf{V}_h$
5: $M_{n+1} \leftarrow \texttt{concat}(M_{n+1}^0, \dots, M_{n+1}^h)$
6: $M_{n+1} \leftarrow M_{n+1} \mathbf{W}_M^T$
   **Output:** $M_{n+1}$

---

action. Each modality is encoded using modality-specific encoders (Algorithm 1): $\tilde{R}_t = \texttt{Encoder}_R(R_t)$, $\tilde{o}_t = \texttt{Encoder}_o(o_t)$, $\tilde{a}_t = \texttt{Encoder}_a(a_t)$. The encoded sequence is split into $N = T//K$ non-overlapping segments $S_n$ of length $K$. Thus, the effective context is $K_{\text{eff}} = N \times K$, well beyond standard attention limits. Each segment is prepended and appended with memory embeddings $M_n \in \mathbb{R}^{m \times d}$, where $m$ is the number of memory tokens and $d$ the embedding dimension: $\tilde{S}_n = \texttt{concat}(M_n, S_n, M_n) \in \mathbb{R}^{(3K+2m) \times d}$ Each segment is then processed by the transformer: $\hat{a}_n, M_{n+1} = \texttt{Transformer}(\tilde{S}_n)$ The output $M_{n+1}$ is then refined via MRV before being passed to the next segment.

Naively forwarding memory embeddings leads to error accumulation or overwriting of relevant information. To address this, we introduce the **Memory Retention Valve (MRV)**, a cross-attention module that filters new memory tokens through the lens of the previous ones (Algorithm 2):

$$\texttt{MRV}(M_n, M_{n+1}) = \texttt{FFN}\left(\texttt{MultiHead}(\text{Query} = M_n, \text{Key} = M_{n+1}, \text{Value} = M_{n+1})\right) \quad (1)$$

This mechanism allows $M_n$ to control what to retain or overwrite when updating to $M_{n+1}$. Unlike static recurrence, it preserves sparse, long-range information. RATE overcomes DT's limits by

132 extending context with recurrence, preserving early cues via MRV, and retaining key events in sparse
133 settings. As a result, RATE solves tasks where DT fails, generalizes beyond training, and remains
134 competitive on standard MDPs.

135 **Attention pattern analysis.** Figure 2 compares attention maps of RATE and DT on a T-Maze
136 sequence. DT (right) attends only within a fixed window, focusing on recent tokens while losing
137 early cues like $o_0$. RATE (left) segments the input and uses memory tokens to propagate information
138 across segments. These tokens retain access to $o_0$ even in later segments, demonstrating RATE's
139 ability to model long-range dependencies beyond the context window through structured memory.

## 3.1 Preservation Properties of MRV

141 We formalize the intuition that the *cross-attention–based* MRV prevents catastrophic overwriting of
142 memory by preserving alignment between consecutive memory states. All vectors are row-vectors.
143 We use $\|\cdot\|_F$ for the Frobenius norm and $\|\cdot\|_2$ for the $\ell_2$ norm.

144 Let $M_n \in \mathbb{R}^{m \times d}$ and $\tilde{M}_{n+1} \in \mathbb{R}^{m \times d}$ denote the incoming and updated memory embeddings at
145 segment $n$, where $m$ is the number of memory tokens and $d$ is the model dimension. We assume that
146 each row $i$ of $M_n$ is $\ell_2$-normalized: $\|M_{n,i}\|_2 = 1$ . The MRV computes the next memory state as:

147 $Q = M_n W_Q, K = \tilde{M}_{n+1} W_K, V = \tilde{M}_{n+1} W_V, A = \texttt{softmax}\left(\frac{QK^\top}{\sqrt{d}}\right), M_{n+1} = AVW_M.$

148 $\alpha$**-alignment condition.** The memory embeddings are said to satisfy $\alpha$-*alignment* if there exists a
149 constant $\alpha \in (0, 1]$ such that for every row $M_{n,i}$, there exists a row $V_j$ for which: $\langle V_j W_M, M_{n,i} \rangle \geq$
150 $\alpha$. This implies that the angle between $V_j W_M$ and $M_{n,i}$ is at most $\arccos \alpha$. Empirically, this
151 condition holds in trained models, as the transformer tends to preserve useful memory content and
152 avoids orthogonal rotations between segments.

153 **Theorem 1** (**On memory loss bounds**). *Let each memory row be $\ell_2$-normalized, the $\alpha$-alignment*
154 *condition hold, and $A = \mathit{softmax}\left(\frac{QK^\top}{\sqrt{d}}\right)$ be the MRV attention matrix. Then:*

$$\|M_{n+1} - M_n\|_F \leq \sqrt{2\left(1 - \frac{\alpha}{m}\right)} \cdot \|M_n\|_F, \quad \|M_{n+1}\|_F \geq \left(1 - \sqrt{2\left(1 - \frac{\alpha}{m}\right)}\right) \cdot \|M_n\|_F. \quad (2)$$

155 *In words: at least a $\left(1 - \sqrt{2\left(1 - \frac{\alpha}{m}\right)}\right)$ fraction of the initial memory is guaranteed to be preserved*
156 *after a single MRV update* (2) *(right), and the memory loss is upper bounded by* (2) *(left).*

157 *Proof.* Since each row of the attention matrix $A$ is a probability distribution, we have $\sum_j A_{ij} = 1$
158 for every $i$. By the pigeonhole principle, there exists an index $j^*$ such that $A_{ij^*} \geq \frac{1}{m}$.

159 By assumption, for each $M_{n,i}$ there exists a $V_j$ such that $\langle V_j W_M, M_{n,i} \rangle \geq \alpha$. In particular, this
160 holds for $j^*$: $\langle V_{j^*} W_M, M_{n,i} \rangle \geq \alpha$. Using the MRV definition $M_{n+1,i} = \sum_j A_{ij} V_j W_M$, we write:

$$\langle M_{n+1,i}, M_{n,i} \rangle = \sum_j A_{ij} \langle V_j W_M, M_{n,i} \rangle \geq A_{ij^*} \langle V_{j^*} W_M, M_{n,i} \rangle \geq \frac{\alpha}{m}. \quad (3)$$

161 Let $\theta_i$ be the angle between $M_{n+1,i}$ and $M_{n,i}$. Since both vectors are $\ell_2$-normalized, we have:
162 $\cos \theta_i = \frac{\langle M_{n+1,i}, M_{n,i} \rangle}{\|M_{n+1,i}\|_2 \cdot \|M_{n,i}\|_2} \geq \frac{\alpha}{m}$. Using the identity $\|u - v\|_2^2 = 2(1 - \cos\theta)$ for unit vectors:
163 $\|M_{n+1,i} - M_{n,i}\|_2^2 \leq 2\left(1 - \frac{\alpha}{m}\right)$, thus $\|M_{n+1,i} - M_{n,i}\|_2 \leq \sqrt{2\left(1 - \frac{\alpha}{m}\right)}$. Summing over all mem-
164 ory tokens and applying the previous bound: $\|M_{n+1} - M_n\|_F^2 = \sum_{i=1}^m \|M_{n+1,i} - M_{n,i}\|_2^2 \leq$
165 $2m\left(1 - \frac{\alpha}{m}\right)$, which simplifies to: $\|M_{n+1} - M_n\|_F \leq \sqrt{2m\left(1 - \frac{\alpha}{m}\right)}$. Consequently, since
166 $\|M_n\|_F = \sqrt{m}$ due to row normalization, we conclude: $\|M_{n+1} - M_n\|_F \leq \sqrt{2\left(1 - \frac{\alpha}{m}\right)} \cdot \|M_n\|_F$.

167 We now derive the lower bound (2) (left) using the reverse triangle inequality. For any matrices
168 $M_{n+1}, M_n \in \mathbb{R}^{m \times d}$, we have: $\|M_{n+1}\|_F \geq \|M_n\|_F - \|M_{n+1} - M_n\|_F$. Substituting the upper
169 bound from (2) (right): $\|M_{n+1} - M_n\|_F \leq \sqrt{2\left(1 - \frac{\alpha}{m}\right)} \cdot \|M_n\|_F$, we obtain: $\|M_{n+1}\|_F \geq$
170 $\left(1 - \sqrt{2\left(1 - \frac{\alpha}{m}\right)}\right) \cdot \|M_n\|_F$, which completes the proof of (2). $\square$

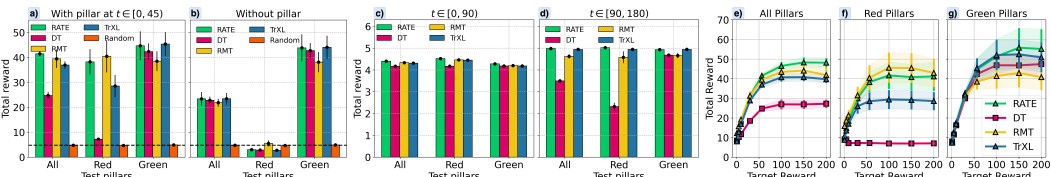

Figure 3: Comparison of RATE with transformer baselines (DT, RMT, TrXL) on ViZDoom-Two-Colors trained on the first $T_{train} = 90$ steps of the episode: with (**a**) and without (**b**) pillar in the first 45 steps of the episode; calculated at environment steps $0 - 89$ (**c**) and $90 - 179$ (**d**) with pillar in the first 45 steps; depending on the return-to-go (**e, f, g**). Episode timeout – 2100 steps.

## 4  Experimental Evaluation

We designed our experiments to achieve two main goals: (a) to showcase the strengths of the RATE model in memory-intensive environments (T-Maze, ViZDoom-Two-Colors, Memory Maze, Minigrid-Memory, POPGym), and (b) to assess its effectiveness in standard MDPs, demonstrating its versatility across domains.

**Baselines.** To evaluate the performance of RATE, we compare it against a diverse set of baselines spanning several categories: **transformer-based models** including *Decision Transformer* (DT) [10], *Recurrent Memory Transformer* (RMT) [7] and *Transformer-XL* (TrXL) [13] specially adapted by us for offline

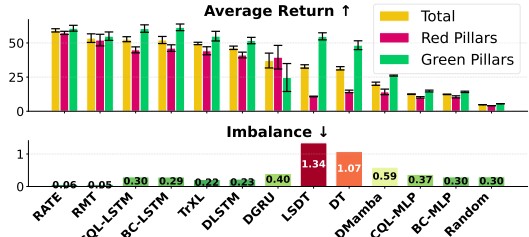

Figure 4: ViZDoom-Two-Colors results with $T_{train}$=150. The top plot shows average return across all episodes (yellow), and separately for red (red) and green (green) pillars. The bottom plot shows the imbalance metric—absolute difference between red and green performance. Lower imbalance indicates more consistent behavior and is as important as average return.

RL, and *Long-Short Decision Transformer* (LSDT) [52]; **classic baselines** such as *Behavior Cloning with* an MLP backbone (BC-MLP) and *Conservative Q-Learning* [26] with an MLP backbone (CQL-MLP); **recurrent models** including *Behavior Cloning with an LSTM backbone* [21] (BC-LSTM), *CQL with LSTM* (CQL-LSTM), *Decision LSTM* (DLSTM) [45], and its *GRU-based variant* [12] (DGRU); and a **state space model** baseline, *Decision Mamba* (DMamba) [35].

**Memory-intensive environments.** We evaluate RATE in tasks that require agents to retain information over time Figure 9; full details are in Appendix C. **ViZDoom-Two-Colors**: the agent must recall a briefly visible pillar color to collect matching items; **T-Maze**: a cue at the start indicates the correct turn at the end, testing sparse long-term memory; **Minigrid-Memory**: like T-Maze, but the clue must be located first, combining memory and credit assignment [34]; **Memory Maze**: the agent searches for objects matching a changing target color, requiring spatial memory; **POPGym**: a suite of 46 partially observable tasks [32] designed to probe different aspects of memory.

### 4.1  Experimental Results

**ViZDoom-Two-Colors.** Figure 4 shows training with $T_{train}$=150 and inference up to 2100 steps, where the pillar disappears at step 90. RATE achieves the highest return and lowest imbalance between the red and green pillars, indicating strong and consistent memory use. Figure 3 further tests transformer models trained with $T_{train} = 90$ on their ability to retain early cues. With the pillar present (a), RATE again yields the highest and most stable return. DT

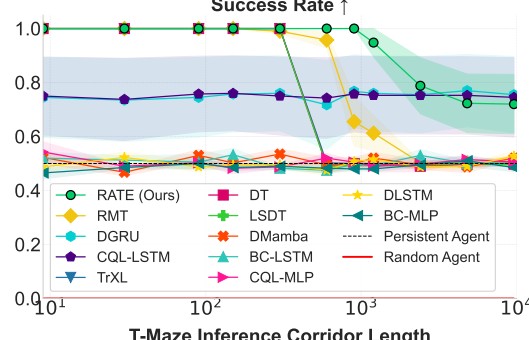

Figure 5: T-Maze generalization task.

and TrXL underperform and show a higher imbalance. Removing the pillar (b) degrades all models, confirming reliance on the initial cue. DT's unchanged performance across (a) and (b) highlights its failure to leverage long-term dependencies.

This limitation is clearer in Figure 3 (c, d), which separates performance within and beyond the 90-step context. DT's return drops by nearly 50% in red-pillar episodes once the cue leaves the

window, while memory models (RATE, RMT, TrXL) remain stable, demonstrating their ability to retain and use information over long horizons.

Figure 3 (e, f, g) shows model performance across target reward levels. RATE consistently outperforms all baselines overall (e), and this advantage is even clearer when separating red (f) and green (g) pillar episodes. While other models show large disparities, RATE maintains stable performance across both conditions, demonstrating effective use of initial cues and validating the strength of its memory architecture.

**T-Maze.** Figure 5 shows the model generalization in Passive T-Maze as inference length grows from 9 to 9600 steps. All models were trained on episodes up to 900 steps; extrapolation beyond this requires long-horizon generalization. RATE achieves 100% success across all in-distribution lengths and performs well even at 9600-step inference, corresponding to trajectories of $3 \times 9600 = 28800$ tokens due to the $(R, o, a)$ triplets. This highlights RATE's ability to retain and leverage sparse cues over extremely long horizons. Other transformers (e.g., DT, LSDT) match RATE on training-length sequences but degrade sharply beyond. DT collapses to $\sim 50\%$ even at moderate lengths due to its lack of memory. Memory-augmented models like RMT generalize slightly further but deteriorate. TrXL performs similarly to DT, suggesting hidden-state caching alone is insufficient for long-range recall of sparse information. RNNs and SSMs (e.g., BC-LSTM, DMamba) show flat curves and fail to learn from sparse long sequences.

RATE both interpolates within training and extrapolates well beyond, a key strength for solving sparse POMDPs. Notably, poor performance of some memory baselines in Figure 5 is due to difficulty modeling long sequences during training, not just generalization failure: even for $T_{\text{val}} \leq T_{\text{train}}$, they may fail. However, when trained on shorter sequences, some models learn generalizable behaviors. Figure 6 visualizes inference performance for RATE (top), DT (middle), and BC-LSTM (bottom) across training/validation lengths. The black dashed line separates *in-distribution* ($T_{\text{val}} \leq T_{\text{train}}$) from *out-of-distribution* ($T_{\text{val}} > T_{\text{train}}$). From Figure 6 (bottom), BC-LSTM generalizes well when trained on short sequences ($\leq 150$), but degrades as training lengths grow, reaching $\sim 0.5$ when trained on $T \geq 600$, likely due to vanishing gradients or limited capacity [37, 49]. DT (Figure 6 (middle)) handles long training sequences via attention, but fails on longer validation sequences due to fixed context. In contrast, RATE (Figure 6 (top)) maintains high success across all validation lengths, enabled by its combination of attention and recurrent memory, which overcomes the limitations of both DT and RNNs.

**Minigrid-Memory.** Figure 7 presents average returns on Minigrid-Memory, where all models were trained on grids of fixed size $41 \times 41$ and evaluated on a wide range of unseen grid sizes from $11 \times 11$ to $501 \times 501$. RATE achieves consistently high performance across the entire spectrum, demonstrating both strong interpolation and extrapolation capabilities. While TrXL also performs well on average, its variance is notably higher, indicating sensitivity to grid scale.

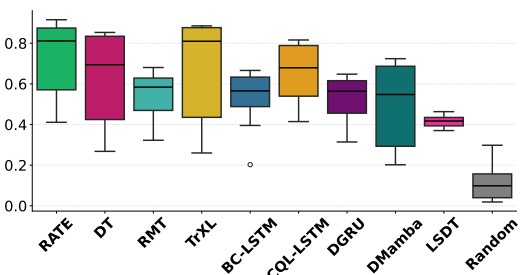

Figure 7: Minigrid-Memory generalization task.

Figure 6: Heatmaps of success rates on T-Maze tasks. The black dashed line separates *in-distribution inference* (with $T_{val} \leq T_{train}$) from *out-of-distribution inference* (with $T_{val} > T_{train}$). Results for other baselines can be found in Appendix, Figure 11.

Table 1: Average return $\pm$ SEM in the Memory Maze ($9 \times 9$) environment (ep. length: 1000 steps).

| Method | Random | BC-LSTM | CQL-LSTM | DT | RMT | TrXL | RATE |
|--------|--------|---------|----------|-----|-----|------|------|
| **Return** | $0.00 \pm 0.00$ | $4.75 \pm 0.15$ | $0.19 \pm 0.02$ | $6.83 \pm 0.51$ | $7.27 \pm 0.21$ | $7.12 \pm 0.24$ | $\mathbf{7.64} \pm 0.41$ |

**Memory Maze.** Table 1 presents results on the Memory Maze task. RATE achieves higher average episode returns by effectively capturing implicit structure, such as maze layout. For reference, the dataset's average return is 4.69. All models were trained on 90-step trajectory subsequences, while full episodes span 1000 steps.

Table 2: Aggregated average returns on 46 POPGym tasks, split into memory and reactive subsets.

| Tasks | Rand. | BC-MLP | DT | BC-LSTM | RATE |
|---|---|---|---|---|---|
| All (46) | -12.2 | -6.8 | 5.8 | 9.0 | **9.5** |
| Memory (31) | -14.6 | -11.9 | -3.5 | -0.2 | **0.5** |
| Reactive (15) | 2.3 | 5.1 | **9.3** | 9.1 | 9.1 |

**POPGym.** To further assess generalization and memory capabilities, we evaluated models on all 46 tasks from the POPGym benchmark suite [32], which covers a wide range of partially observable RL scenarios. The benchmark is split into 31 *memory puzzle tasks* and 15 *reactive POMDP tasks*. Table 2 reports average normalized scores across all tasks and subsets. RATE achieves the highest overall score ($9.54$), outperforming all baselines. On the challenging memory tasks, RATE maintains a positive average score ($0.45$), while all other models fall below zero – indicating a consistent failure to exploit long-term dependencies. Notably, DT scores $-3.49$ and BC-MLP drops to $-11.91$, highlighting the limitations of both context-limited transformers and non-recurrent policies.

On reactive tasks, all models perform better, but the gap between memory-based and non-memory models narrows. RATE, DT, and BC-LSTM show almost the same results, suggesting that the greatest performance gains from RATE's memory mechanisms occur on memory puzzle tasks. For simpler reactive POMDPs, lightweight memory mechanisms appear sufficient. These results also underscore RATE's ability to generalize across both puzzle and reactive settings, confirming that its memory architecture does not hinder performance in simpler tasks while offering clear benefits in those with temporal dependencies. More details are provided in Appendix, Table 9.

Table 3: Normalized scores on MuJoCo tasks from the D4RL benchmark [16]. Although RATE is designed for memory-intensive environments, it **performs competitively** – and often surpasses – methods tailored for standard MDP control. **Top-1** and **Top-2** results are highlighted.

| Dataset | Environment | CQL | DT | TAP | TT | DMamba | MambaDM | RATE (ours) |
|---|---|---|---|---|---|---|---|---|
| ME | HalfCheetah | 91.6 | 86.8±1.3 | 91.8±0.8 | 95.0±0.2 | 91.9±0.6 | 86.5±1.2 | 87.4±0.1 |
| ME | Hopper | 105.4 | 107.6±1.8 | 105.5±1.7 | 110.0±2.7 | 111.1±0.3 | 110.5±0.3 | 112.5±0.2 |
| ME | Walker2d | 108.8 | 108.1±0.2 | 107.4±0.9 | 101.9±6.8 | 108.3±0.5 | 108.8±0.1 | 108.7±0.5 |
| M | HalfCheetah | 44.4 | 42.6±0.1 | 45.0±0.1 | 46.9±0.4 | 42.8±0.1 | 42.8±0.1 | 43.5±0.3 |
| M | Hopper | 58.0 | 67.6±1.0 | 63.4±1.4 | 61.1±3.6 | 83.5±12.5 | 85.7±7.8 | 77.4±1.4 |
| M | Walker2d | 72.5 | 74.0±1.4 | 64.9±2.1 | 79.0±2.8 | 78.2±0.6 | 78.2±0.6 | 80.7±0.7 |
| MR | HalfCheetah | 45.5 | 36.6±0.8 | 40.8±0.6 | 41.9±2.5 | 39.6±0.1 | 39.1±0.1 | 39.0±0.6 |
| MR | Hopper | 95.0 | 82.7±7.0 | 87.3±2.3 | 91.5±3.6 | 82.6±4.6 | 86.1±2.5 | 83.7±8.2 |
| MR | Walker2d | 77.2 | 66.6±3.0 | 66.8±3.1 | 82.6±6.9 | 70.9±4.3 | 73.4±2.6 | 73.7±1.4 |
| | Average | 77.6 | 74.7 | 74.8 | 78.9 | 78.8 | 79.0 | 78.5 |

**Atari and MuJoCo.** We evaluate RATE on standard RL benchmarks: Atari games and MuJoCo control tasks (Table 3, Table 4). For comparison, we include results from recent state-of-the-art methods: Decision Mamba (DMamba) [35], Mamba as Decision Maker (MambaDM) [8], Conservative Q-Learning (CQL) [26], Trajectory Transformer (TT) [22], and TAP [23], as reported in their original papers. Results show that RATE matches or outperforms specialized offline RL algorithms across both benchmarks. Combined with its strong performance on memory-intensive tasks, this highlights RATE's versatility as a general-purpose offline RL model.

See Appendix E for full training details and Table 10 for the evaluation protocol.

# 5 Ablation Study

We conduct a comprehensive ablation study to assess the contributions of individual components and architectural choices in RATE, structured around three key research questions.

1. *How do different components of RATE influence performance on memory tasks?* (RQ1)

2. *What is the upper-bound results RATE can achieve with access to perfect memory?* (RQ2)

3. *What role does the MRV play, and which configuration is most effective?* (RQ3)

Further ablations exploring key transformer parameters, memory tokens number, and sequence segmentation strategies are provided in Appendix F and Appendix G.

Table 4: Raw scores on Atari games. RATE outperforms DT in 3 out of 4 environments.

| Environment | CQL | BC | DT | DMamba | MambaDM | RATE (Ours) |
|---|---|---|---|---|---|---|
| Breakout | 62.5 | 42.8 | $76.9_{\pm 27.3}$ | $70.6_{\pm 9.3}$ | $\mathbf{106.9}_{\pm 5.8}$ | $\mathbf{111.0}_{\pm 2.9}$ |
| Qbert | **14013.2** | 2862.0 | $2215.8_{\pm 1523.7}$ | $5786.0_{\pm 1295.2}$ | $10052.5_{\pm 1116.5}$ | $\mathbf{12486.9}_{\pm 280.4}$ |
| SeaQuest | 782.2 | **992.1** | $\mathbf{1129.3}_{\pm 189.0}$ | $992.1_{\pm 57.7}$ | $\mathbf{1286.0}_{\pm 42.0}$ | $1037.9_{\pm 53.7}$ |
| Pong | **18.8** | 6.4 | $\mathbf{17.1}_{\pm 2.9}$ | $1.6_{\pm 15.3}$ | $\mathbf{18.4}_{\pm 0.8}$ | $\mathbf{18.8}_{\pm 0.3}$ |

**RQ1: Impact of RATE components.** To assess the contribution of individual memory mechanisms in RATE, we conducted inference-time ablations by replacing memory components with random noise. In T-Maze ($K = 30$, $N = 3$ segments), corrupting the memory embeddings $M$ caused a sharp drop in performance to a 50% success rate (see Figure 8, right). Notably, the agent still reached the decision point but failed to choose the correct direction—indicating that while the model retained its navigation policy, it lost access to the initial cue. This implies that memory embeddings serve as dedicated storage for task-relevant information, while transformer layers encode general behavioral patterns. In ViZDoom-Two-Colors (see Figure 8, left), we further disentangled the roles of memory components by selectively adding noise to memory embeddings and cached hidden states. The results revealed that performance was more sensitive to the corruption of cached hidden states, underscoring their importance in environments with continuous rewards and extended dependency chains. Together, these findings suggest a division of roles: memory embeddings are essential for sparse, discrete decision points (e.g., in T-Maze), whereas cached representations are more critical in dense, continuous-feedback environments like ViZDoom-Two-Colors.

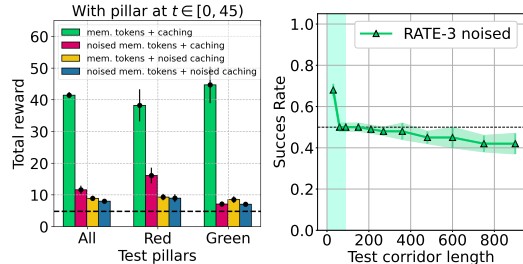

Figure 8: Effect of memory corruption on RATE at inference. **(left)** ViZDoom: performance drops when memory tokens or cached states are noised. **(right)** T-Maze: SR degrades when memory embeddings are corrupted.

**RQ2: Performance upper-bound estimate.** To estimate the upper-bound performance achievable by RATE, we introduce *OracleDT* – a variant of Decision Transformer augmented with perfect prior knowledge about the environment. Specifically, OracleDT receives an additional input vector $v \in \mathbb{R}^{1 \times \texttt{d\_model}}$ prepended and appended to the context sequence, i.e., $S' = \texttt{concat}(v, S, v)$. This vector encodes one bit of environment-critical information known in advance. In T-Maze, $v$ represents the initial clue ($v_i = 0$ if left, $v_i = 1$ if right); in ViZDoom-Two-Colors, it encodes the pillar color ($v_i = 0$ for red, $v_i = 1$ for green). This setup mirrors a context augmented with perfectly trained memory embeddings, i.e., $\texttt{concat}(M, S, M)$, where $M$ encodes all relevant information. As a result, OracleDT provides an empirical upper bound on achievable performance when key information is available explicitly. In such settings, we expect the relation $R[\text{OracleDT}] \geq R[\text{RATE}] \geq R[\text{DT}]$ to hold (see Table 5). Since this privileged information is not generally accessible during training, OracleDT is not a viable baseline but serves as a useful reference. The gap between OracleDT and RATE quantifies the effectiveness of RATE's memory mechanisms in autonomously discovering, storing, and utilizing task-relevant information.

Table 5: Performance comparison between DT, RATE, and OracleDT. OracleDT is an oracle-informed variant used solely to approximate the upper bound and is not a feasible baseline.

| T-Maze | | | |
|---|---|---|---|
| **Success Rate** | **OracleDT** | **DT** | **RATE** |
| $T = 90$ | $\mathbf{1.00}_{\pm 0.00}$ | $\mathbf{1.00}_{\pm 0.00}$ | $\mathbf{1.00}_{\pm 0.00}$ |
| $T = 480$ | $\mathbf{1.00}_{\pm 0.00}$ | $0.50_{\pm 0.00}$ | $0.90_{\pm 0.07}$ |
| $T = 900$ | $\mathbf{1.00}_{\pm 0.00}$ | $0.50_{\pm 0.00}$ | $0.90_{\pm 0.07}$ |
| **ViZDoom-Two-Colors** | | | |
| **Total Reward** | $\mathbf{56.5}_{\pm 0.8}$ | $24.8_{\pm 1.4}$ | $41.5_{\pm 1.0}$ |
| **Red Pillars** | $\mathbf{55.3}_{\pm 1.6}$ | $7.2_{\pm 0.4}$ | $38.2_{\pm 5.1}$ |
| **Green Pillars** | $\mathbf{57.2}_{\pm 0.5}$ | $42.3_{\pm 3.3}$ | $44.7_{\pm 5.8}$ |

**RQ 3. Memory Retention Valve scheme ablation.** In the T-Maze environment, we observed that without MRV, RATE's performance deteriorates on long corridors ($L \gg K$), eventually reaching SR $= 50\%$ (see Table 6). This degradation occurs because critical information to be remembered goes into memory embeddings when processing the first segment of the sequence, and then it must be retrieved when making decisions on the last segment. At the same time, due to the recurrent structure of the architecture, memory embeddings continue to be updated during the processing of intermediate segments when no new information needs to be memorized, causing important information from

memory embeddings to leak out. To address this information loss, we introduced the **Memory Retention Valve (MRV)** and evaluated five variants: **MRV-CA-1**: Cross-attention mechanism where updated embeddings ($M_{n+1}$) query incoming ones ($M_n$); **MRV-CA-2**: Reversed variant where incoming embeddings ($M_n$) query updated ones ($M_{n+1}$); **MRV-G**: Gating mechanism inspired by GTrXL [36]; **MRV-GRU**: GRU-based [12] memory processing with hidden states; **MRV-LSTM**: LSTM-based [21] memory processing with cell states.

Table 6: Ablation of MRV configurations in T-Maze ($K_{\text{eff}}=30\times5=150$). Baseline without MRV is marked †. Default: **MRV-CA-2**.

| Model | 150 | 360 | 600 | 900 |
|---|---|---|---|---|
| w/o MRV† | 1.00 ±0.00 | 0.66 ±0.08 | 0.65 ±0.07 | 0.61 ±0.07 |
| **MRV-CA-2** | 1.00 ±0.00 | 0.95 ±0.05 | 0.90 ±0.07 | 0.90 ±0.07 |
| MRV-G | 0.86 ±0.07 | 0.77 ±0.08 | 0.66 ±0.07 | 0.65 ±0.08 |
| MRV-GRU | 0.99 ±0.01 | 0.74 ±0.07 | 0.56 ±0.11 | 0.55 ±0.12 |
| MRV-LSTM | 0.85 ±0.06 | 0.64 ±0.10 | 0.51 ±0.11 | 0.47 ±0.11 |
| MRV-CA-1 | 0.51 ±0.01 | 0.51 ±0.01 | 0.49 ±0.02 | 0.49 ±0.01 |

Among all tested configurations, MRV-CA-2 demonstrated best performance (see Table 6). This cross-attention scheme uses incoming memory tokens ($M_n$) as queries and updated tokens ($M_{n+1}$) as keys and values. This configuration, referred to simply as MRV throughout the paper, effectively controls information flow through memory. By allowing the model to selectively update its memory based on the relevance of new information, it prevents loss of important context over long sequences.

## 6 Related Work

**Transformers in RL.** Transformers have been applied to online [36, 27, 33, 43, 28], offline [10, 22, 52], and model-based RL [9]. While prior work often relies on compact observations or known dynamics [29, 23], RATE targets long-horizon credit assignment and memory challenges in partially observable environments, using DT [10] as a baseline. A recent extension, *Long-Short Decision Transformer* (LSDT) [52], augments DT with two context windows but still lacks an explicit, learnable memory. Retrieval-augmented variants, e.g. RA-DT [43], index external trajectories for in-context planning, while Fast and Forgetful Memory (FFM) [33] and Stable Hadamard Memory (SHM) [28] explore lightweight recurrent memory slots with improved stability.

**RNNs in RL.** Recurrent models like LSTM [21] and GRU [12] have long been used for memory in RL. DLSTM [45] replaces transformers with LSTM to support sequential decision-making. However, RNNs often struggle with long-term dependencies, especially in sparse-reward settings [34].

**SSMs in RL.** SSMs such as S4 [19] and Mamba [18] offer efficient alternatives to attention, showing strong offline RL results [3, 35, 8]. These models rely on linear dynamics and their ability to handle memory-intensive generalization remains unclear.

**Memory-Augmented Transformers.** Memory extensions like T*ransformer-XL* [13], *Compressive Transformer* [41], and RMT [7] improve context handling via caching or compression. RATE builds on these ideas by combining token-level memory, hidden-state caching, and a novel MRV gate.

## 7 Limitations

While RATE is tailored for long-horizon, memory-intensive tasks, its complexity may be unnecessary in fully observable or short-term settings where simpler recurrent models suffice. Nonetheless, RATE matches or exceeds their performance across all tasks. Future work may explore adaptive variants that scale memory based on task complexity.

## 8 Conclusion

We propose the **Recurrent Action Transformer with Memory** (**RATE**), a transformer-based architecture for offline RL that combines attention with recurrence for long-horizon decision-making. RATE integrates memory embeddings, hidden state caching, and a **Memory Retention Valve** (**MRV**) to selectively retain critical information across segments. RATE achieves state-of-the-art results on memory-intensive tasks such as T-Maze, Minigrid-Memory, ViZDoom-Two-Colors, Memory Maze, and POPGym, generalizing up to 9600-step sequences and outperforming both recurrent and transformer baselines. Theoretical analysis shows that MRV guarantees lower-bounded memory preservation across updates, and ablation studies confirm its importance for long-horizon stability. Despite its memory focus, RATE also performs competitively on standard benchmarks like Atari and MuJoCo, demonstrating broad versatility. These results establish RATE as a unified, general-purpose offline RL model that excels across both short and long temporal contexts.

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

# Table of Contents

## A    Discussion: Are RNNs Still Better for Memory?

Our experiments provide a systematic comparison between recurrent and transformer-based archi-
tectures in memory-intensive tasks. When trained on short sequences, recurrent models such as
BC-LSTM perform competitively. For example, in the T-Maze environment, BC-LSTM achieves
perfect success rates when trained on sequences up to 150 steps, effectively capturing short-term
dependencies via its internal state dynamics.

However, this advantage quickly fades as training sequences grow longer. Increasing the training
horizon from 150 to 600 steps causes BC-LSTM's performance to collapse to a 50% success rate
across all inference lengths—even those shorter than the training context—indicating difficulty with
gradient stability and information retention over long spans (Figure 6). In contrast, RATE maintains
consistently high performance under the same conditions, demonstrating stronger scalability with
sequence length. RATE generalizes robustly to inference horizons up to 9600 steps (28,800 tokens),
reflecting the effectiveness of its hybrid memory design. The architecture combines token-based
recurrence with gated memory updates via the Memory Retention Valve (MRV), enabling reliable
propagation of sparse information across long temporal distances.

These findings extend to more complex environments. In ViZDoom-Two-Colors and Memory Maze
(Figure 4, Table 1), RATE significantly outperforms BC-LSTM. In ViZDoom, RATE maintains
balanced performance across red and green cues, whereas BC-LSTM exhibits instability and higher
variance. In Memory Maze, RATE achieves substantially higher returns, benefiting from its capacity
to encode and retrieve spatial-temporal patterns over long episodes.

In conclusion, while RNNs remain effective for short-range temporal dependencies, their performance
degrades in long-horizon, sparse-reward, and generalization-critical settings. RATE bridges this gap

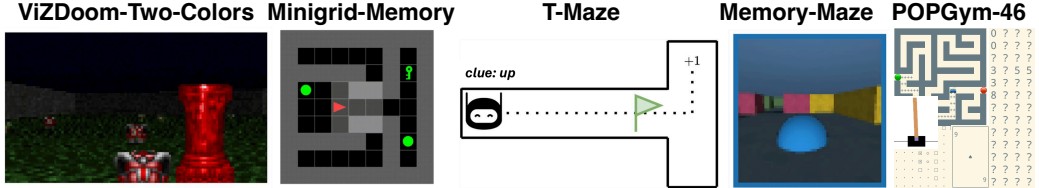

| ViZDoom-Two-Colors | Minigrid-Memory | T-Maze | Memory-Maze | POPGym-46 |

Figure 9: Memory-intensive environments used to evaluate RATE memory mechanisms.

by integrating attention with recurrence, offering a scalable and robust memory solution. These results underscore the architectural promise of combining transformer attention with recurrent dynamics for long-term tasks in RL.

# B  Decision Transformer

Decision Transformer (DT) [10] is an algorithm for offline RL that reduces the RL task to a sequence modeling task. In DT, the scheme of which is presented in Algorithm 3, the trajectory $\tau$ is not divided into segments as in RATE. Instead, random fragments of length $K$ are sampled from the trajectory, since originally this architecture was designed to work only with MDP. The predicted actions $\hat{a}$ are sampled autoregressively.

---

**Algorithm 3** Decision Transformer

**Require**: $R \in \mathbb{R}^{1 \times T}, o \in \mathbb{R}^{d_o \times T}, a \in \mathbb{R}^{1 \times T}$

1: $\tilde{R} \in \mathbb{R}^{T \times d} \leftarrow \texttt{Encoder}_R(R)$
   $\tilde{o} \in \mathbb{R}^{T \times d} \leftarrow \texttt{Encoder}_o(o)$
   $\tilde{a} \in \mathbb{R}^{T \times d} \leftarrow \texttt{Encoder}_a(a)$
2: $\tau_{0..T} \leftarrow \{(\tilde{R}_0, \tilde{o}_0, \tilde{a}_0), \ldots, (\tilde{R}_T, \tilde{o}_T, \tilde{a}_T)\}$
3: $n = \texttt{random}(0, T - K)$
4: $\hat{a}_n \leftarrow \texttt{Transformer}(\tau_{n..n+K})$

**Output**: $\hat{a}_n \rightarrow \mathcal{L}(a_n, \hat{a}_n)$

---

# C  Environments

## C.1  Memory-intensive environments

In this section, we provide an extended description of the environments used in this paper, as well as the methodology used to collect the trajectories. Table 7 summarizes the observations type, rewards type, and actions type for each of the environments considered in this paper.

### C.1.1  ViZDoom-Two-Colors

We used a modified ViZDoom-Two-Colors environment from [46] to assess the model's memory abilities. The agent initially having 100 hit points (HP) is placed in a room without inner walls filled with acid. At each step in the environment, the agent loses a fixed amount of health ($10/32$ HP per step). In the center of the environment, there is a pillar of either green or red color, which disappears after 45 environment steps. Throughout the environment, objects of two colors (green and red) are generated. When the agent interacts with an object of the same color as the pillar, it gains an increase in health of $+25$ and a reward of $+1$. When the agent interacts with an object of the opposite color, it loses a similar amount of health. The agent receives an additional reward of $+0.02$ for each step it survives. The episode ends when the agent has zero health. Thus, the agent needs to remember the color of the pillar to select items of the correct color, even if the pillar is out of sight or has disappeared. The agent does not receive information about its current health or rewards, as these observations essentially convey the same information as the color of the pillar but persist beyond step 45.

We collected a dataset of 5000 trajectories of 90 steps in length using a trained A2C [4] agent (an agent trained with a non-disappearing pillar). The average reward for these 90 steps is $4.46$. When collecting trajectories, to ensure that the agent saw the pillar before it disappeared, the agent always appeared facing the pillar in the same place – midway between the pillar and the nearest wall. In order to successfully complete this task, the agent needs to remember the color of the pillar. This environment tests the long-term memory mechanism, since the agent needs to retain information about the pillar for a time much longer than the pillar has been in the environment. Using only short-term

Table 7: Description of observations and reward functions for the considered environments.

| Environment | Obs. Type | Rew. Type | Act. Space | Obs. Details |
|---|---|---|---|---|
| ViZDoom-Two-Colors | Image | Continuous | Discrete | First-person view |
| T-Maze | Vector | Sparse & Discrete | Discrete | Low-dimensional vector |
| Memory Maze | Image | Sparse & Discrete | Discrete | First-person view |
| Minigrid-Memory | Image | Sparse | Discrete | $3 \times 3$ grid centered on agent |
| POPGym | Vector/Image | Discrete/Continuous | Discrete/Continuous | Vector or 2D grid |
| Action Assoc. Retrieval | Vector | Sparse & Discrete | Discrete | Symbolic vector input |
| Atari | Image | Sparse & Discrete | Discrete | Full game screen |
| MuJoCo | Vector | Continuous | Continuous | Low-dimensional state vector |

memory and, for example, collecting the next item of the same color as the previous collected item, it will not be possible for the agent to survive for a long time, as this policy is extremely unstable. This is due to the fact that in the training dataset the agent occasionally makes a mistake and picks up an object of the opposite color. Thus, irrelevant information about the desired color may enter the transformer context and the agent will start collecting items of an opposite color, which will quickly lead to a failure.

### C.1.2   T-Maze

To investigate agent's long-term memory on very long environments (the inference trajectory length is much longer than the effective context length $K_{eff}$) we used a modified version of the T-Maze environment [34]. The agent's objective in this environment is to navigate from the beginning of the T-shaped maze to the junction and choose the correct direction, based on a signal given at the beginning of the trajectory using four possible actions $a \in \{left, up, right, down\}$. This signal, represented as the $clue$ variable and equals to zero everywhere except the first observation, dictates whether the agent should turn up ($clue = 1$) or down ($clue = -1$). Additionally, a constraint on the episode duration $T = L + 2$, where the maximum duration is determined by the length of the corridor $L$ to the junction, adds complexity to the problem. To address this, a binary flag, represented as the $flag$ variable, which is equal to $1$ one step before the junction and $0$ otherwise, indicating the arrival of the agent at the junction, is included in the observation vector. Additionally, a noise channel is added to the observation vector, with random integer values from the set $\{-1, 0, +1\}$. The observation vector is thus defined as $o = [y, clue, flag, noise]$, where $y$ represents the vertical coordinate. The reward $r$ is given only at the end of the episode and depends on the correctness of the agent's turn at the junction, being $1$ for a correct turn and $0$ otherwise. This formulation deviates from the traditional Passive T-Maze environment [34] (different observations and reward functions) and presents a more intricate set of conditions for the agent to navigate and learn within the given time constraint.

The dataset consists of 2000 of trajectories for each segment of length 30 (i.e. 6000 trajectories for the $K_{eff} = 3 \times 30 = 90$) and consists only of successful episodes. An artificial oracle with a priori information about the environment was used to generate the dataset.

### C.1.3   Memory Maze

In this first-person view 3D environment [38], the agent appears in a randomly generated maze containing several objects of different colors at random locations. The agent's task is to find an object of the same color in the maze as the outline around its observation image. After the agent finds an object of the desired color and steps on it, the color of the outline changes and the agent must find another object. The agent receives a $+1$ reward for stepping on the correct object. Otherwise, it receives no reward. The duration of an episode is a fixed number and is equal to 1000. Thus, the agent's task is to find as many objects of the desired color as possible in a limited time. The agent's effectiveness in this environment depends on its ability to memorize the structure of the maze and the location of objects in it in order to find the desired objects faster. Using the Dreamer model [20] to collect dataset of 5000 trajectories only achieved an average award of 4.7 per episode, i.e., a rather sparse dataset.

### C.1.4 Minigrid-Memory

Minigrid-Memory [11] is a 2D grid environment designed to test an agent's long-term memory and credit-assignment [34]. The environment map is a T-shaped maze with a small room with an object inside it at the beginning of the corridor. The agent appears at a random coordinate in the corridor. The agent's task is to reach the room with the object and memorize it, then reach the junction at the end of the maze and make a turn in the direction where the same object is located as in the room at the beginning of the maze. A reward $r = 1 - 0.9 \times \frac{t}{T}$ is given for success, and $0$ for failure. The episode ends after any agent turns at a junction or after a limited amount of time (95 steps) has elapsed. The agent's observations are limited to a $3 \times 3$ size frame. 10000 trajectories with grid size 41x41 were collected using PPO [44] with Transformer-XL (TrXL) [39] with a context length equal to the maximum episode duration.

### C.1.5 POPGym

POPGym [32] is a benchmark suite consisting of 46 diverse partially observable environments designed to isolate different aspects of memory use and generalization in reinforcement learning. The tasks include both short-horizon reactive scenarios and long-horizon memory puzzles that require the agent to remember information across extended delays or infer hidden states from past observations. The environments vary in observation modality (image vs. vector), reward sparsity, and temporal dependencies. For our dataset, we followed the original POPGym evaluation protocol and used a PPO [44] agent with a GRU [12] backbone (PPO-GRU), which showed the best performance in the original benchmark. We collected trajectories using this policy for all 46 environments. The collected dataset reflects the diverse difficulty and memory requirements of the benchmark and serves as a challenging testbed for evaluating general-purpose memory architectures like RATE.

## C.2 Standard benchmarks

### C.2.1 Atari games

For the Atari game environments [5], we used the same dataset as in DT, namely the DQN replay dataset with grayscale state images [2]. This dataset contains $500$ thousand of the $50$ million steps of an online DQN [31] agent for each game. We use the following set of games: SeaQuest, Breakout, Pong and Qbert.

### C.2.2 MuJoCo.

Despite the fact that memory is not required in decision making in control environments like MuJoCo [16], we conducted additional experiments in this environment to compare with DT. For the continuous control tasks, we selected a standard MuJoCo locomotion environment and a set of trajectories from the D4RL benchmark [16]. Since we chose DT and TAP as the main models for comparison on this data, we focused on the environments used in both works (HalfCheetah, Hopper, and Walker). We used three

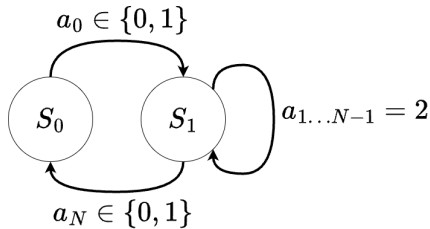

Figure 10: Action Associative Retrieval.

different dataset settings: 1) **Medium** – 1 million timesteps generated by a "medium" policy that achieves about a third of the score of an expert policy; 2) **Medium-Replay** – the replay buffer of an agent trained with the performance of a medium policy (about 200k–400k timesteps in our environments); 3) **Medium-Expert** – 1 million timesteps generated by the medium policy concatenated with 1 million timesteps generated by an expert policy. The scores for the MuJoCo experiments are normalized such that 100 represents an expert policy, following the benchmark protocol outlined in [16]. The performance metrics for Conservative Q-Learning (CQL) and Trajectory Autoencoding Planner (TAP) are reported from the TAP paper [23], and for DT from the DT paper [10], as they use the same dataset and evaluation protocol.

## D Action Associative Retrieval

As shown in Figure 6, DT has a SR $= 50\%$ for inference at corridor lengths longer than the transformer context length. This is due to the fact that even a DT trained on balanced data has a slight bias in

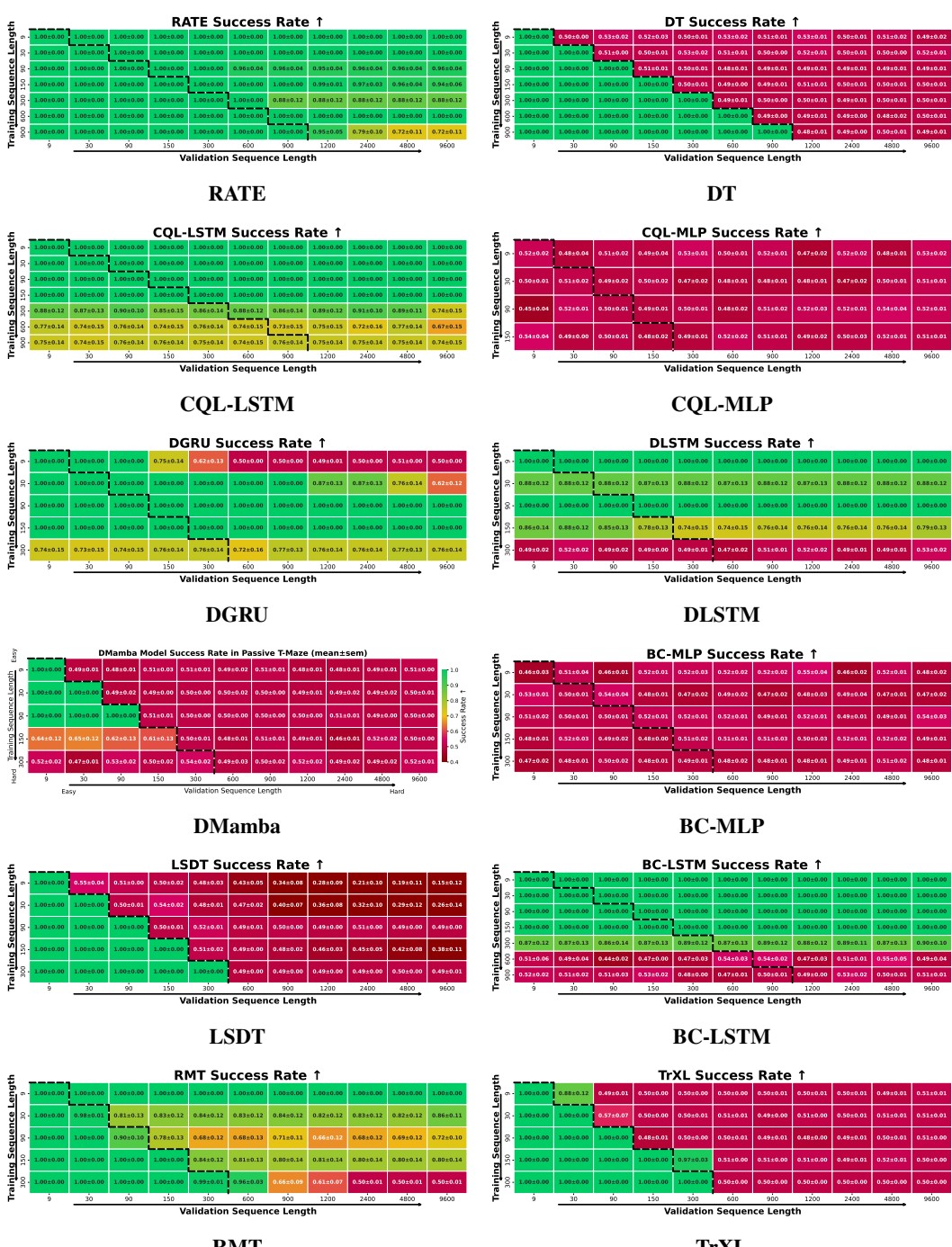

Figure 11: Results for all models in the T-Maze generalization task.

the predicted probability towards one of the two required actions, which leads to the fact that when $t > K$ the agent constantly produces only one action: up or down. In turn, the presence of memory in the agent allows us to combat this problem.

To check how the agent's performance changes during training, we design an **Action Associative Retrieval** (**AAR**) Figure 10 environment.

There are two states in this environment: $S_0$ and $S_1$. The agent appears in state $S_0$ and by performing the action $a_0 \in \{0, 1\}$ moves to state $S_1$. Next, the agent must take $N - 2$ steps to move from state

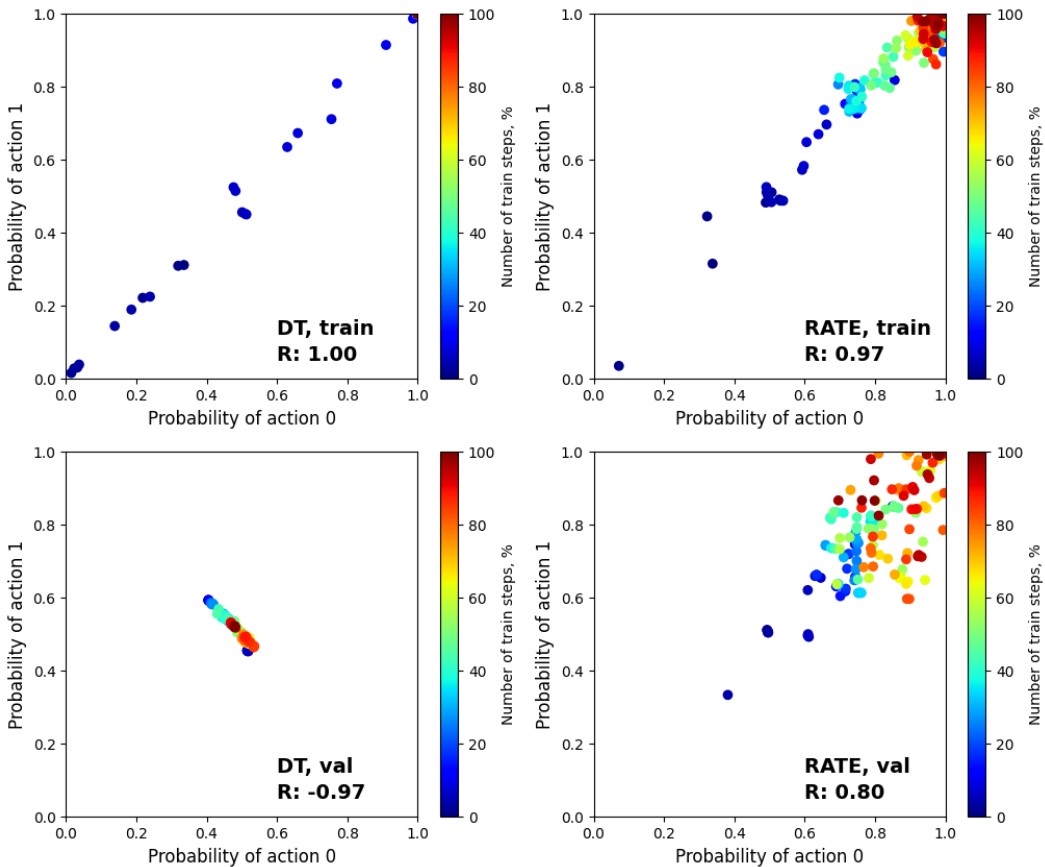

Figure 12: Experimental results with RATE and DT in the AAR environment. The graphs show the 10-runs average results of training on trajectories of length $T = 90$ and validation on trajectories of length $T = 180$, for RATE with $K_{eff} = 3 \times 30 = 90$ and for DT with $K = 90$.

$S_1$ to state $S_1$ by performing action $a = 2$ (no op.). At the end of the episode, the agent must perform the same action that moved it from state $S_0$ to state $S_1$ in order to move from state $S_1$ to state $S_0$. Thus, the action $a \in \{0, 1, 2\}$. Agent observations $o = [state, flag, noise]$, where $state \in \{0, 1\}$ is the index of the current state, $flag \in \{0, 1\}$ is a flag equal to 1 in case the next step requires returning to the initial state and equal to 0 otherwise, $noise \in \{-1, 0, +1\}$ is the noise channel. The agent receives a $+1$ reward if it returns to the initial state $S_0$ by performing the action that took it out from the $S_0$ to the $S_1$, and $-1$ in other cases. The training dataset consists of oracle-generated 6000 trajectories with positive reward.

More formally, we can talk about the presence of memory in an agent when solving AAR (T-Maze-like) tasks under the condition that:

$$\forall t > K : \frac{1}{N_0} \sum_{i=1}^{N_0} p_i(a_t = a^0 | a_0 = a^0) + \frac{1}{N_1} \sum_{i=1}^{N_1} p_i(a_t = a^1 | a_0 = a^1) > 1 \qquad (4)$$

This condition means that if the agent has memory, the sum of the average conditional probabilities over all experiments will be greater than one, i.e., these probabilities are independent of each other. Provided that the sum of these probabilities is less than or equal to one, the agent will choose at best the same target action in most experiments, even if another action is required.

where $a^0, a^1 \in \mathcal{A}$ – two mutually exclusive actions leading to a reward; $t$ is the step at which the final action is required; $N_0, N_1$ are the number of experiments in environments where target action $a_t = a^0$ and $a_t = a^1$, respectively.

In the results Figure 12, the first $1\%$ of training steps was removed because it corresponds to the beginning of the training and is unrepresentative. Blue dots correspond to the beginning of training, red dots to the end of training. As can be seen from Figure 12, during training, the probabilities $p_i(a_t = a^0 | a_0 = a^0)$ and $p_i(a_t = a^1 | a_0 = a^1)$ on the training trajectories have a strong positive correlation ($R_{train}^{DT} = 1.00$ and $R_{train}^{RATE} = 0.97$), where $R$ – correlation coefficient. This indicates that within-context (effective context) DT and RATE models are able to predict both $a^0$ and $a^1$ actions equally well.

At the same time, during validation, for the RATE model this pattern is preserved – the red points corresponding to the probabilities of choosing actions $a^0$ and $a^1$ are in the upper right part of the graph, positive correlation persists ($R_{val}^{RATE} = 0.80$). On the other hand, in the DT case, the cluster of red dots is skewed toward choosing action $a^1$ and action $a^0$ with equal probabilities equal to $0.5$. Thus, in sum, these probabilities are less or equal to one, as evidenced by a strong negative correlation ($R_{val}^{DT} = -0.97$). The results confirm the inability of DT to generalize on trajectories whose lengths exceed the context length and the ability of RATE to handle such tasks.

# E   Training

This section provides additional details on the training process of the baselines considered in the paper. We treated the inclusion of the feed-forward network (FFN) block in RATE's transformer decoder as a hyperparameter, as RATE performed slightly better without FFN in some environments. In contrast, other transformer-based baselines were trained with the standard transformer decoder including FFN.

## E.1   ViZDoom-Two-Colors

Since the pillar disappears at time $t=45$, all trajectories span from $t=0$ to $t=90$ to ensure that the cue remains available during training. In this setting, we compare DT with context length $K=90$ to RATE, RMT, and TrXL models using $K=30$ and $N=3$ segments. Thus, RATE processes sequences of the same total length $K_{\text{eff}}=N \times K=90$ but accesses only $K=30$ tokens at a time. Additionally, we ran experiments with $N=3$, $K=50$, and $T=150$ to validate model robustness under longer and more complex configurations.

## E.2   Passive T-Maze

We trained models on sequences of length $T_{\text{train}} \in \{9, 30, 90, 150, 300, 600, 900\}$ and evaluated them on $T_{\text{val}} \in \{9, 30, 90, 150, 300, 600, 900, 1200, 2400, 4800, 9600\}$. For RATE, each sequence was split into $N = 3$ segments, yielding a context length of $K = T_{\text{train}}/3$. All training trajectories started from $t = 0$, ensuring the cue was always included. In what follows, we adopt the notation **MODEL-N**, where $N = 3$ indicates segmentation into three recurrent blocks (e.g., RATE-3 is trained on full sequences of length $T = 90$ with $K = 30$). This convention is used throughout the ablation studies.

## E.3   Memory Maze

To train RATE, DT, RMT, and TrXL on Memory Maze, we used the same approach as for ViZDoom-Two-Colors environment, but instead of using fixed trajectories starting at $t = 0$, we sampled consecutive 90-step subsequences from the original 1000-step trajectories. Each subsequence was sampled with a stride of 90 steps, resulting in approximately 11 training sequences per original trajectory. As in the ViZDoom-Two-Colors case, training for DT was performed with a context length of $K = 90$ and for RATE, RMT, and TrXL with a context length of $K = 30$ and number of segments $N = 3$, i.e., effective context length $K_{eff} = N \times K = 3 \times 30 = 90$.

## E.4   Minigrid-Memory

To train baselines in this environment, we used only mazes of fixed size $41 \times 41$, ensuring a consistent corridor length during training. For evaluation, models were validated on mazes ranging from $11 \times 11$ to $501 \times 501$, where corridor lengths vary within each grid, enabling assessment of both interpolation and extrapolation capabilities. All training trajectories used an episode timeout of 96 steps, while

validation trajectories across all maze sizes used a longer timeout of 500 steps. As in T-Maze, each trajectory began at $t = 0$, ensuring the cue was always observed. During training, RATE used a context length of $K = 30$ with $N = 3$ segments, while other baselines (except RMT and TrXL) used $K = 90$.

### E.5 POPGym Suite

POPGym [32] comprises 46 tasks of varying memory complexity, including both memory puzzles and reactive POMDPs. Since episode lengths vary widely across tasks – from as short as 12 steps to as long as 1000 – we ensured a consistent and fair memory evaluation for RATE by setting the context length $K = T/3$ and using $N = 3$ segments for every environment, where $T$ denotes the maximum episode length of each task. This uniform configuration allowed RATE to process full trajectories with recurrent segmentation, ensuring its memory capacity was equally tested across tasks of different lengths and difficulties.

### E.6 Atari and MuJoCo

When training RATE on Atari games and MuJoCo control tasks, sequences of length $T = 90$ (Atari) and $T = 60$ (MuJoCo) were sampled randomly from the original trajectories in the dataset. These trajectories were then divided into $N = 3$ segments of length $K = 30$ (Atari) and $K = 20$ (MuJoCo), forming an effective context of length $K_{eff} = N \times K = 90$ (60 for MuJoCo).

For Atari, we used the identical experimental design described in the DT paper [10]. It is worth noting that we presented raw scores for Atari, rather than gamer-normalized scores as described in the DT paper. Table 4 shows the results for Atari environments. RATE outperforms DT significantly in environments like Breakout and Qbert. We attribute this to the observation that, although these environments do not explicitly demand memory, intricate dynamics from the past exert a greater influence on agent behavior than in environments such as SeaQuest. Actions executed in the past notably alter the present state of the environment in Breakout and Qbert, whereas in SeaQuest, such actions hold little significance. For instance, the emergence of enemies and divers in SeaQuest is entirely independent of the agent's prior actions.

For MuJoCo, our findings suggest that the conventional strategy of utilizing return is not suitable for our segment-based scheme. The issue arises during the trajectory, where the agent's return persistently diminishes. However, the true value of the agent's state at the onset and conclusion of the episode could remain unchanged, provided the agent's policy performs consistently well. To rectify this discrepancy, we propose a novel evaluation strategy for MuJoCo tasks. In this approach, each segment commences with the maximum return, simulating the scenario where the agent initiates the trajectory anew. This method effectively mitigates the aforementioned issue, enhancing the accuracy of our evaluation process. Our MuJoCo experiments in Table 3 show that this benefits performance significantly for some environments. Thus, using RATE allowed us to obtain the best metrics for MuJoCo in 4/9 cases compared to the other baselines. RATE also outperforms DT in 9/9 tasks.

## F Additional ablation studies

To determine the optimal hyperparameters associated with memory mechanisms, additional ablation studies were performed in ViZDoom-Two-Colors and T-Maze environments, and the results are presented in Figure 14 and Figure 13 (right). From the ablation studies results, it was found that for environments like ViZDoom-Two-Colors with continuous reward signal and image observations, the best results can be obtained using number of cached memory tokens `mem_len` $= (K \times 3 + 2 \times$ `num_mem_tokens`$) \times N$, where $K$ – context length and $N$ – number of segments.

On the other hand, for environments with sparse events like T-Maze, it has been found that using caching of hidden states of previous tokens (`mem_len` $> 0$) prevents remembering important information.

### F.1 Additional ViZDoom-Two-Colors ablation

The effect of combining of memory tokens with noise is shown in Figure 13 (left). The noise was applied as a convex combination: `memory_tokens` $= (1 - \alpha) \times$ `memory_tokens` $+ \alpha \times$ `noise`.

Table 8: RATE hyperparameters for different experiments. ‡ – Leaky ReLU used in Atari.Pong. The listed hyperparameters for ViZDoom-Two-Colors and T-Maze correspond to the experiments with $T_{\text{train}} = 150$, while for POPGym, they reflect the settings used in the POPGym-Concentration task.

| Hyperparameter | ViZDoom2C | Memory Maze | T-Maze | Minigrid-Memory | POPGym | Atari | MuJoCo |
|---|---|---|---|---|---|---|---|
| *Memory-specific parameters* | | | | | | | |
| Number of memory tokens | 15 | 15 | 10 | 10 | 30 | 15 | 5 |
| Number of cached tokens | 100 | 360 | 0 | 180 | 100 | 360 | 60 |
| Number of MRV heads | 2 | 0 | 2 | 4 | 2 | 1 | 1 |
| MRV activation | ReLU | ReLU | ReLU | ReLU | ReLU | ReLU‡ | ReLU |
| *Transformer architecture* | | | | | | | |
| Number of layers | 6 | 6 | 8 | 4 | 10 | 6 | 3 |
| Number of attention heads | 8 | 8 | 8 | 4 | 2 | 8 | 1 |
| Embedding dimension | 64 | 64 | 64 | 128 | 32 | 128 | 128 |
| Context length $K$ | 50 | 30 | 50 | 30 | 18 | 30 | 20 |
| Number of segments | 3 | 3 | 3 | 3 | 3 | 3 | 3 |
| Skip dec FFN | False | True | True | False | True | True | True |
| *Regularization* | | | | | | | |
| Hidden dropout | 0.2 | 0.5 | 0.2 | 0.3 | 0.1 | 0.2 | 0.2 |
| Attention dropout | 0.05 | 0.2 | 0.1 | 0.1 | 0.05 | 0.05 | 0.05 |
| Weight decay | 0.001 | 0.1 | 0.001 | 0.001 | 0.001 | 0.1 | 0.1 |
| *Training configuration* | | | | | | | |
| Max epochs | 150 | 80 | 200 | 500 | 200 | 10 | 10 |
| Batch size | 128 | 64 | 64 | 64 | 32 | 128 | 4096 |
| Loss function | CE | CE | CE | CE | CE | CE | MSE |
| Optimizer | AdamW | AdamW | AdamW | AdamW | AdamW | AdamW | AdamW |
| Learning rate | 3e-4 | 3e-4 | 1e-4 | 1e-4 | 3e-4 | 3e-4 | 6e-5 |
| Grad norm clip | 5.0 | 1.0 | 1.0 | 5.0 | 5.0 | 1.0 | 1.0 |
| Cosine decay | False | True | False | False | False | True | False |
| Linear warmup | True | True | True | True | True | True | True |
| $(\beta_1, \beta_2)$ | (0.9, 0.999) | (0.9, 0.95) | (0.9, 0.999) | (0.9, 0.999) | (0.9, 0.999) | (0.9, 0.95) | (0.9, 0.95) |

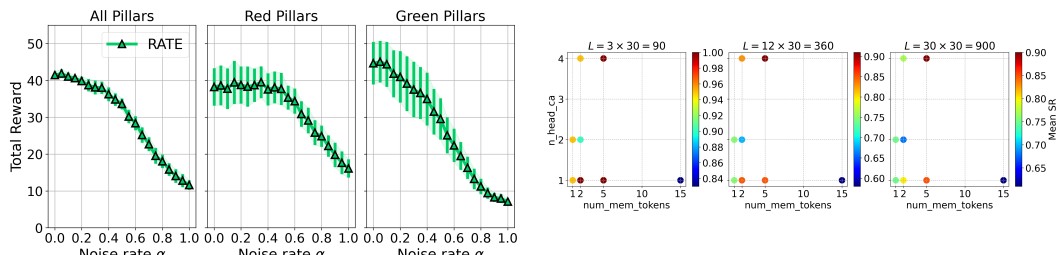

Figure 13: (**left**) Investigating the RATE memory tokens noise effect in the ViZDoom-Two-Colors. (**right**) Results of RATE-3 (trained on corridor lengths $\leq 90$) ablation studies in the T-Maze environment. `n_head_ca` – number of MRV attention heads, `num_mem_tokens` – number of memory tokens.

With unchanged caching of hidden states from previous steps at growth of the noise parameter $\alpha$, at first there is a decrease of performance at inference on green pillars (up to $\alpha = 0.5$), and only then a decrease of performance at inference on red pillars. This phenomenon can be explained by the fact that memory embeddings is trained to record mostly information about red pillars, which helps to combat bias in the training data.

## F.2 Curriculum Learning

Since in the T-Maze environment, the number of actions at the junction relates to the number of actions when moving straight along the corridor as $\frac{1}{L}$ and tends to $0$ as $L$ increases, there is a significant imbalance in the agent's action distribution, which can cause problems when performing rare class (turning actions) prediction. Theoretically, this situation can be remedied through curriculum learning.

Curriculum learning (CL) is a technique in which a model is trained on examples of increasing difficulty. In this approach, the model is first trained on the set of trajectories $Q_1 = q_1$ of length $K \times 1$, then the trained model is re-trained on the set of trajectories $Q_2 = q_1 \cup q_2$, where the set

Table 9: Performance on POPGym tasks (mean±sem over three runs, 100 seeds each).

| Environment | RATE | DT | Random | BC-MLP | BC-LSTM | Dataset Average Return |
|---|---|---|---|---|---|---|
| AutoencodeEasy-v0 | $-0.29 \pm 0.00$ | $-0.47 \pm 0.00$ | $-0.50 \pm 0.00$ | $-0.47 \pm 0.00$ | $-0.32 \pm 0.00$ | $-0.26$ |
| AutoencodeMedium-v0 | $-0.47 \pm 0.00$ | $-0.49 \pm 0.00$ | $-0.50 \pm 0.00$ | $-0.49 \pm 0.00$ | $-0.47 \pm 0.00$ | $-0.48$ |
| AutoencodeHard-v0 | $-0.46 \pm 0.00$ | $-0.49 \pm 0.00$ | $-0.50 \pm 0.01$ | $-0.50 \pm 0.00$ | $-0.44 \pm 0.00$ | $-0.43$ |
| BattleshipEasy-v0 | $-0.81 \pm 0.02$ | $-0.93 \pm 0.03$ | $-0.46 \pm 0.01$ | $-1.00 \pm 0.00$ | $-0.49 \pm 0.01$ | $-0.35$ |
| BattleshipMedium-v0 | $-0.91 \pm 0.02$ | $-0.91 \pm 0.03$ | $-0.39 \pm 0.00$ | $-1.00 \pm 0.00$ | $-0.81 \pm 0.02$ | $-0.43$ |
| BattleshipHard-v0 | $-0.92 \pm 0.01$ | $-0.97 \pm 0.01$ | $-0.41 \pm 0.00$ | $-1.00 \pm 0.00$ | $-0.67 \pm 0.01$ | $-0.40$ |
| ConcentrationEasy-v0 | $-0.06 \pm 0.02$ | $-0.05 \pm 0.01$ | $-0.19 \pm 0.01$ | $-0.92 \pm 0.00$ | $-0.14 \pm 0.00$ | $-0.12$ |
| ConcentrationMedium-v0 | $-0.84 \pm 0.00$ | $-0.84 \pm 0.00$ | $-0.84 \pm 0.00$ | $-0.88 \pm 0.00$ | $-0.84 \pm 0.00$ | $-0.87$ |
| ConcentrationHard-v0 | $-0.25 \pm 0.00$ | $-0.25 \pm 0.01$ | $-0.19 \pm 0.00$ | $-0.92 \pm 0.00$ | $-0.19 \pm 0.01$ | $-0.44$ |
| CountRecallEasy-v0 | $0.07 \pm 0.01$ | $-0.46 \pm 0.01$ | $-0.93 \pm 0.00$ | $-0.92 \pm 0.00$ | $0.05 \pm 0.00$ | $0.22$ |
| CountRecallMedium-v0 | $-0.47 \pm 0.01$ | $-0.75 \pm 0.03$ | $-0.88 \pm 0.00$ | $-0.88 \pm 0.00$ | $-0.47 \pm 0.00$ | $-0.48$ |
| CountRecallHard-v0 | $-0.54 \pm 0.00$ | $-0.81 \pm 0.02$ | $-0.93 \pm 0.00$ | $-0.92 \pm 0.00$ | $-0.56 \pm 0.00$ | $-0.55$ |
| HigherLowerEasy-v0 | $0.50 \pm 0.00$ | $0.50 \pm 0.00$ | $0.00 \pm 0.01$ | $0.47 \pm 0.00$ | $0.50 \pm 0.00$ | $0.51$ |
| HigherLowerMedium-v0 | $0.50 \pm 0.00$ | $0.50 \pm 0.00$ | $-0.01 \pm 0.00$ | $0.49 \pm 0.00$ | $0.50 \pm 0.00$ | $0.49$ |
| HigherLowerHard-v0 | $0.52 \pm 0.00$ | $0.51 \pm 0.00$ | $0.01 \pm 0.01$ | $0.50 \pm 0.00$ | $0.51 \pm 0.01$ | $0.49$ |
| LabyrinthEscapeEasy-v0 | $0.95 \pm 0.00$ | $0.80 \pm 0.01$ | $-0.39 \pm 0.00$ | $0.72 \pm 0.05$ | $0.92 \pm 0.01$ | $0.95$ |
| LabyrinthEscapeMedium-v0 | $-0.81 \pm 0.01$ | $-0.82 \pm 0.01$ | $-0.94 \pm 0.01$ | $-0.89 \pm 0.01$ | $-0.86 \pm 0.00$ | $-0.94$ |
| LabyrinthEscapeHard-v0 | $-0.56 \pm 0.01$ | $-0.67 \pm 0.04$ | $-0.84 \pm 0.04$ | $-0.71 \pm 0.03$ | $-0.69 \pm 0.02$ | $-0.49$ |
| LabyrinthExploreEasy-v0 | $0.95 \pm 0.00$ | $0.88 \pm 0.06$ | $-0.34 \pm 0.01$ | $0.87 \pm 0.01$ | $0.93 \pm 0.00$ | $0.96$ |
| LabyrinthExploreMedium-v0 | $0.79 \pm 0.00$ | $0.77 \pm 0.01$ | $-0.73 \pm 0.00$ | $0.26 \pm 0.01$ | $0.71 \pm 0.01$ | $0.79$ |
| LabyrinthExploreHard-v0 | $0.88 \pm 0.00$ | $0.86 \pm 0.01$ | $-0.61 \pm 0.00$ | $0.45 \pm 0.01$ | $0.82 \pm 0.01$ | $0.87$ |
| MineSweeperEasy-v0 | $0.15 \pm 0.03$ | $-0.33 \pm 0.04$ | $-0.26 \pm 0.03$ | $-0.47 \pm 0.01$ | $0.20 \pm 0.00$ | $0.28$ |
| MineSweeperMedium-v0 | $-0.44 \pm 0.00$ | $-0.40 \pm 0.01$ | $-0.43 \pm 0.00$ | $-0.49 \pm 0.00$ | $-0.35 \pm 0.01$ | $-0.27$ |
| MineSweeperHard-v0 | $-0.20 \pm 0.00$ | $-0.37 \pm 0.02$ | $-0.39 \pm 0.01$ | $-0.48 \pm 0.00$ | $-0.16 \pm 0.00$ | $-0.10$ |
| MultiarmedBanditEasy-v0 | $0.37 \pm 0.01$ | $0.27 \pm 0.01$ | $0.02 \pm 0.00$ | $0.05 \pm 0.00$ | $0.17 \pm 0.02$ | $0.62$ |
| MultiarmedBanditMedium-v0 | $0.22 \pm 0.03$ | $0.27 \pm 0.01$ | $0.01 \pm 0.00$ | $0.01 \pm 0.00$ | $0.17 \pm 0.01$ | $0.43$ |
| MultiarmedBanditHard-v0 | $0.32 \pm 0.01$ | $0.35 \pm 0.01$ | $0.01 \pm 0.00$ | $0.21 \pm 0.01$ | $0.14 \pm 0.00$ | $0.59$ |
| NoisyPositionOnlyCartPoleEasy-v0 | $0.88 \pm 0.03$ | $0.87 \pm 0.02$ | $0.11 \pm 0.00$ | $0.23 \pm 0.00$ | $0.44 \pm 0.01$ | $0.98$ |
| NoisyPositionOnlyCartPoleMedium-v0 | $0.18 \pm 0.01$ | $0.17 \pm 0.01$ | $0.11 \pm 0.00$ | $0.16 \pm 0.00$ | $0.22 \pm 0.01$ | $0.36$ |
| NoisyPositionOnlyCartPoleHard-v0 | $0.33 \pm 0.01$ | $0.34 \pm 0.00$ | $0.12 \pm 0.01$ | $0.18 \pm 0.00$ | $0.25 \pm 0.01$ | $0.57$ |
| NoisyPositionOnlyPendulumEasy-v0 | $0.87 \pm 0.00$ | $0.84 \pm 0.01$ | $0.27 \pm 0.01$ | $0.31 \pm 0.00$ | $0.88 \pm 0.00$ | $0.90$ |
| NoisyPositionOnlyPendulumMedium-v0 | $0.60 \pm 0.01$ | $0.56 \pm 0.01$ | $0.26 \pm 0.00$ | $0.28 \pm 0.00$ | $0.66 \pm 0.00$ | $0.67$ |
| NoisyPositionOnlyPendulumHard-v0 | $0.68 \pm 0.00$ | $0.63 \pm 0.01$ | $0.27 \pm 0.01$ | $0.30 \pm 0.00$ | $0.72 \pm 0.00$ | $0.73$ |
| PositionOnlyCartPoleEasy-v0 | $0.93 \pm 0.03$ | $1.00 \pm 0.00$ | $0.12 \pm 0.00$ | $0.15 \pm 0.00$ | $0.17 \pm 0.00$ | $1.00$ |
| PositionOnlyCartPoleMedium-v0 | $0.05 \pm 0.01$ | $0.03 \pm 0.00$ | $0.04 \pm 0.00$ | $0.05 \pm 0.00$ | $0.06 \pm 0.00$ | $1.00$ |
| PositionOnlyCartPoleHard-v0 | $0.07 \pm 0.00$ | $0.34 \pm 0.08$ | $0.05 \pm 0.00$ | $0.09 \pm 0.00$ | $0.12 \pm 0.00$ | $1.00$ |
| PositionOnlyPendulumEasy-v0 | $0.54 \pm 0.02$ | $0.51 \pm 0.01$ | $0.27 \pm 0.00$ | $0.29 \pm 0.00$ | $0.91 \pm 0.00$ | $0.92$ |
| PositionOnlyPendulumMedium-v0 | $0.47 \pm 0.01$ | $0.49 \pm 0.01$ | $0.26 \pm 0.00$ | $0.28 \pm 0.00$ | $0.82 \pm 0.00$ | $0.82$ |
| PositionOnlyPendulumHard-v0 | $0.49 \pm 0.01$ | $0.55 \pm 0.01$ | $0.26 \pm 0.00$ | $0.30 \pm 0.00$ | $0.89 \pm 0.00$ | $0.88$ |
| RepeatFirstEasy-v0 | $1.00 \pm 0.00$ | $0.45 \pm 0.16$ | $-0.49 \pm 0.01$ | $-0.50 \pm 0.00$ | $1.00 \pm 0.00$ | $1.00$ |
| RepeatFirstMedium-v0 | $0.10 \pm 0.02$ | $0.42 \pm 0.14$ | $-0.50 \pm 0.00$ | $-0.50 \pm 0.00$ | $-0.50 \pm 0.00$ | $0.99$ |
| RepeatFirstHard-v0 | $0.99 \pm 0.01$ | $-0.21 \pm 0.18$ | $-0.50 \pm 0.00$ | $-0.50 \pm 0.00$ | $0.99 \pm 0.01$ | $1.00$ |
| RepeatPreviousEasy-v0 | $1.00 \pm 0.00$ | $1.00 \pm 0.00$ | $-0.49 \pm 0.01$ | $-0.52 \pm 0.00$ | $1.00 \pm 0.00$ | $1.00$ |
| RepeatPreviousMedium-v0 | $-0.46 \pm 0.00$ | $-0.47 \pm 0.00$ | $-0.51 \pm 0.00$ | $-0.48 \pm 0.00$ | $-0.45 \pm 0.00$ | $-0.48$ |
| RepeatPreviousHard-v0 | $-0.38 \pm 0.01$ | $-0.38 \pm 0.00$ | $-0.50 \pm 0.01$ | $-0.50 \pm 0.00$ | $-0.38 \pm 0.00$ | $-0.39$ |
| VelocityOnlyCartPoleEasy-v0 | $1.00 \pm 0.00$ | $1.00 \pm 0.00$ | $0.11 \pm 0.00$ | $0.99 \pm 0.00$ | $1.00 \pm 0.00$ | $1.00$ |
| VelocityOnlyCartPoleMedium-v0 | $1.00 \pm 0.00$ | $0.96 \pm 0.02$ | $0.04 \pm 0.00$ | $0.63 \pm 0.00$ | $1.00 \pm 0.00$ | $0.99$ |
| VelocityOnlyCartPoleHard-v0 | $1.00 \pm 0.00$ | $1.00 \pm 0.00$ | $0.06 \pm 0.00$ | $0.83 \pm 0.01$ | $1.00 \pm 0.00$ | $1.00$ |

$q_2$ is formed by trajectories of length $K \times 2$, and so on (in order of increasing complexity of the trajectories). Thus, for the $N$ segments considered during training, the set $Q_N = \bigcup_{i=1}^{N} q_i$ is used.

In the T-Maze environment, DT, RATE, RMT, and TrXL were trained with and without curriculum learning because this approach theoretically produces better results. However, it is important to note that the T-Maze task is successfully solved by the RATE model without using curriculum learning, and even vice versa – its use slightly degraded performance on long corridors. However, with respect to TrXL, the use of CL yielded slightly better results. The work showed that using CL does not achieve significantly better performance on the T-Maze task. The results of using the CL on the T-Maze environment are presented in Figure 16 (left), and the results of applying noise to memory embeddings to assess its importance are presented in Figure 16 (right).

Table 10: Experimental setup and evaluation metrics across different environments. $N_{runs}$ denotes the number of model runs; $N_{seeds}$ denotes the number of inference episodes with different seeds; sem denotes standard error of the mean, and std denotes standard deviation.

| Environment | Experiment Setup | | Results | |
| --- | --- | --- | --- | --- |
| | $N_{\text{runs}}$ | $N_{\text{seeds}}$ | Metric | Notation |
| *Memory-intensive environments* | | | | |
| ViZDoom-Two-Colors | 6 | 100 | Return | mean±sem |
| T-Maze | 4 | 100 | Success Rate | mean±sem |
| Memory Maze | 3 | 100 | Return | mean±sem |
| Minigrid-Memory | 3 | 100 | Return | mean±sem |
| POPGym | 3 | 100 | Return | mean±sem |
| *Diagnostic environment* | | | | |
| Action Associative Retrieval | 10 | — | Success Rate | mean±sem |

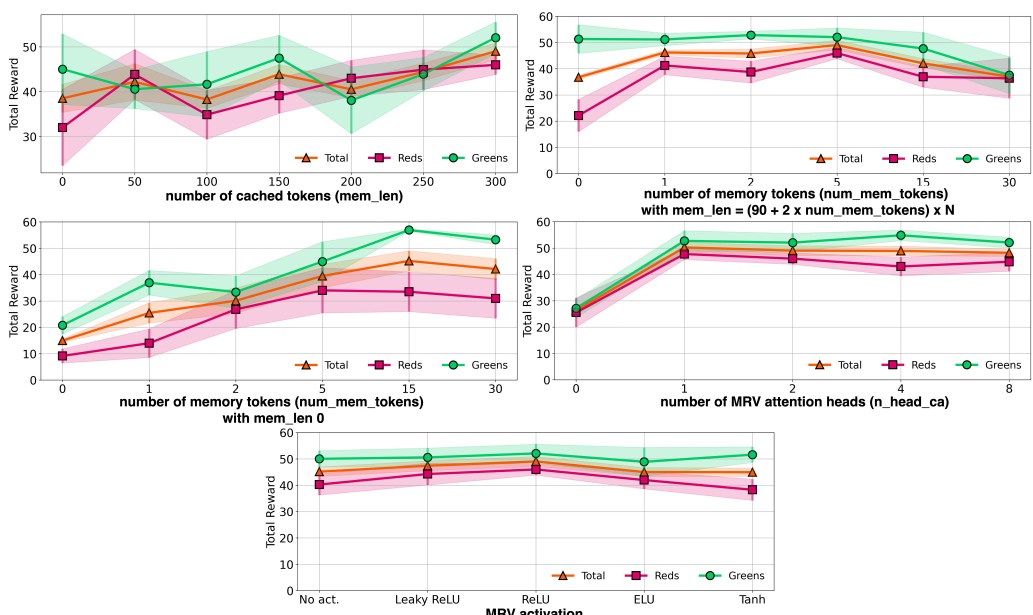

Figure 14: Results of RATE ablation studies in the ViZDoom-Two-Colors environment.

### F.3 Supplemental MRV ablation

One of the options for implementing the memory tokenization gating mechanism was an approach similar to the one proposed in Gated Transforer-XL (GTrXL) [36] work. Thus, the MRV-G scheme was inspired by the gating mechanism from GTrXL and implemented as follows:

$$r = \sigma(M_n W_r + M_{n+1} U_r) \tag{5}$$

$$z = \sigma(M_n W_z + M_{n+1} U_z - \texttt{bias}) \tag{6}$$

$$h = \texttt{tanh}(M_n W_g + (M_{n+1} \times r)U_r) \tag{7}$$

$$\tilde{M}_{n+1} = \sigma(M_n(1 - z) + z \times h) \tag{8}$$

The results of the RATE (trained on corridor lengths of $\leq 150$) inference on the T-Maze environment with these MRV configurations are shown in Figure 17 and in Table 6. The results presented in Figure 17 confirm the high stability of RATE when using cross-attention-based MRV (MRV-CA-2), as well as the model's ability to hold important information in memory embeddings when inference on long tasks.

Table 11: RATE encoders for each part of $(R, o, a)$ triplets. We use an Embedding layer for encoding discrete actions and a Linear layer for continuous ones. ‡ – channels / kernel sizes / padding. For POPGym tasks with grid-based observations (e.g., MineSweeper and Battleship), we encoded the grid using a token dictionary followed by a linear encoder to produce a fixed-length vector. Actions were encoded using an embedding layer for all discrete control tasks, while a linear layer was used for continuous control environments (e.g., PositionOnlyPendulum).

| Environment | Encoder Configuration | | | |
| --- | --- | --- | --- | --- |
| | Return | Observation | Conv. params‡ | Action |
| *Image-based environments* | | | | |
| ViZDoom-Two-Colors | Linear | Conv2D $\times$ 3 | (32, 64, 64) / (8, 4, 3) / 0 | Embedding |
| Memory Maze | Linear | Conv2D $\times$ 3 | (32, 64, 64) / (8, 4, 3) / 2 | Embedding |
| Minigrid-Memory | Linear | Conv2D $\times$ 3 | (32, 64, 64) / (8, 4, 3) / 0 | Embedding |
| Atari | Linear | Conv2D $\times$ 3 | (32, 64, 64) / (8, 4, 3) / 0 | Embedding |
| *Vector-based environments* | | | | |
| T-Maze | Linear | Linear | — | Embedding |
| MuJoCo | Linear | Linear | — | Linear |
| Action Associative Retrieval | Linear | Linear | — | Embedding |
| POPGym | Linear | Linear | — | Embedding / Linear |

## F.4 Ablation on number of segments and segment length

Partitioning the trajectories into fixed-length segments allows the RATE model to train on long trajectories without increasing the context size, which makes the parameters $N$ (the number of segments into which the training trajectories are divided) and $K$ (the context length, i.e., the size of a single segment) critical because they determine the length of the effective context $K_{eff} = K \times N$. Figure 18 presents the results of ablation studies for parameters $N$ and $K$ at fixed $K_{eff} = 90$.

## G  Transformer Ablation Studies

**Transformer core hyperparameters.**  This section presents the results of ablation studies on the main hyperparameters of the RATE transformer. The RATE configuration for the T-Maze environment specified in Table 8 was chosen for the ablation studies. The ablation studies focus on understanding the impact of key hyperparameters by systematically varying one parameter while keeping others constant. The results are shown in Figure 20, Figure 21, and Figure 22.

**Feed-Forward Network.**  For RATE, the inclusion of the decoder feed-forward block is treated as a tunable hyperparameter. In most environments, we disable it, as doing so often leads to better performance Figure 19. However, for ViZDoom-Two-Colors and Minigrid-Memory, we found that retaining the feed-forward block yields slightly improved results, and thus it is enabled in those settings.

## H  Recommendations for Hyperparameter Settings

Transformer-based models require careful hyperparameter tuning, and the addition of memory mechanisms in RATE introduces a few more components. However, **configuring RATE remains largely similar to tuning a standard transformer**. Based on extensive empirical evaluation, we provide the following **practical guidelines** to simplify the setup process.

**Step-by-step configuration:**

1. **Segment setup.** Divide each trajectory into $N = 3$ segments. For a trajectory of length $T$, set the context length to $K = T//3$.

2. **Memory configuration.** Use the following default parameters for RATE's memory mechanisms:

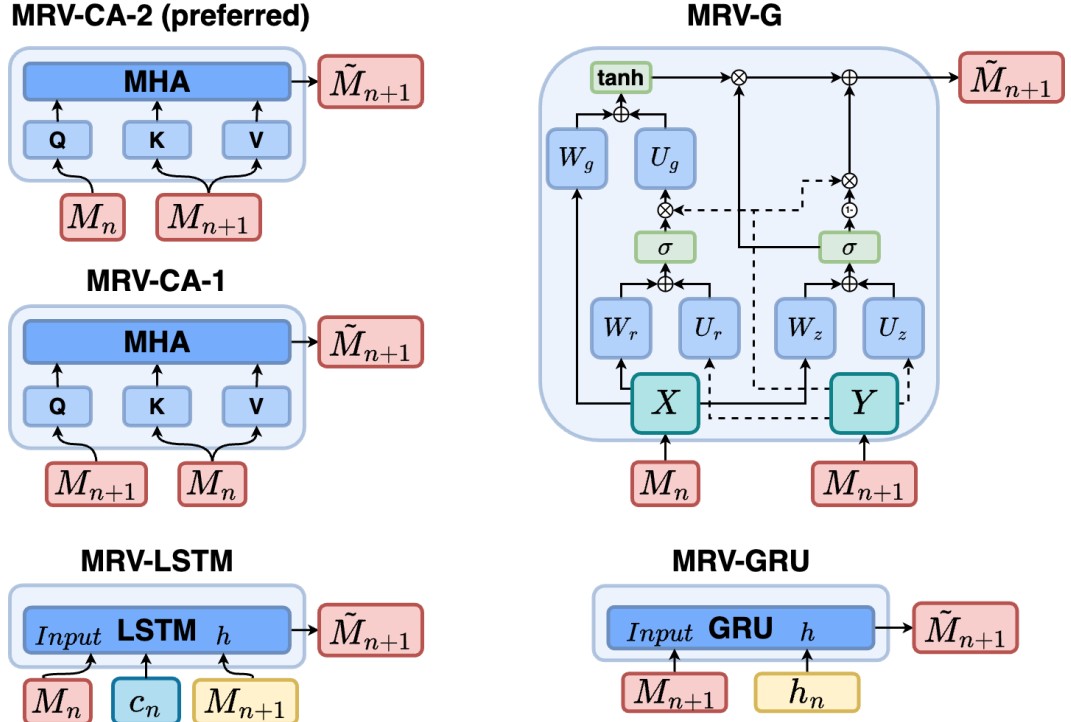

Figure 15: Memory Retention Valve configurations used in the ablation study. **MRV-CA-2**: cross-attention-based MRV which uses an attention mechanism to control the updating of memory embeddings and which is used in the work as the main mechanism. **MRV-CA-1**: uses the same mechanism as MRV-CA-2 but the updated memory embeddings $M_{n+1}$ are fed to Query, and the incoming memory embeddings $M_n$ are fed to Key and Value. **MRV-G**: gated MRV which uses a gating mechanism similar to the one used in Gated Transformer-XL [36]. **MRV-GRU**: uses a GRU [12] block to process updated memory embeddings with hidden states. **MRV-LSTM**: uses a LSTM [21] block to process updated memory embeddings with cached states.

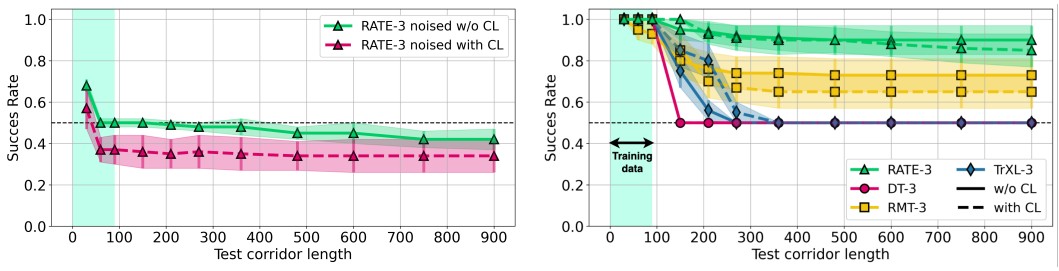

Figure 16: (**left**). Results with and without the use of curriculum learning and (**right**) results of replacing RATE memory tokens with white noise at inference in T-Maze.

- $\texttt{num\_mem\_tokens} = 5$
- $\texttt{n\_head\_ca} = 1$
- $\texttt{mrv\_act} = \mathrm{ReLU}$
- $\texttt{mem\_len} =$
  - $(3 \times K + 2 \times \texttt{num\_mem\_tokens}) \times N$ for dense reward environments (e.g., ViZDoom-Two-Colors, Minigrid-Memory)
  - $0$ for sparse reward environments (e.g., T-Maze)

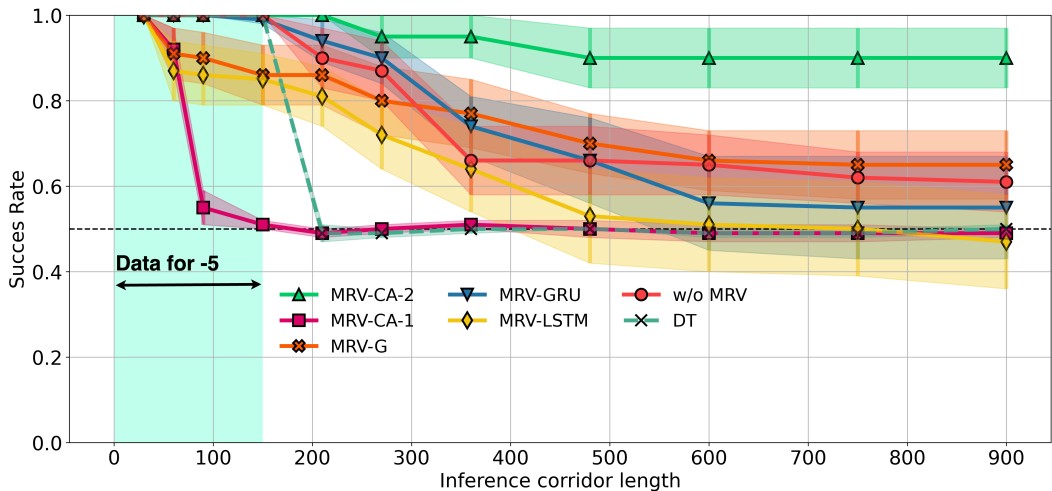

Figure 17: Results of RATE inference with different MRV configurations on the T-Maze environment. Training was performed with the number of segments $N = 5$ and context length $K = 30$, i.e. on trajectories of length $\leq 150$. MRV-CA-2 is the final MRV configuration that is used throughout the work and is designated as MRV.

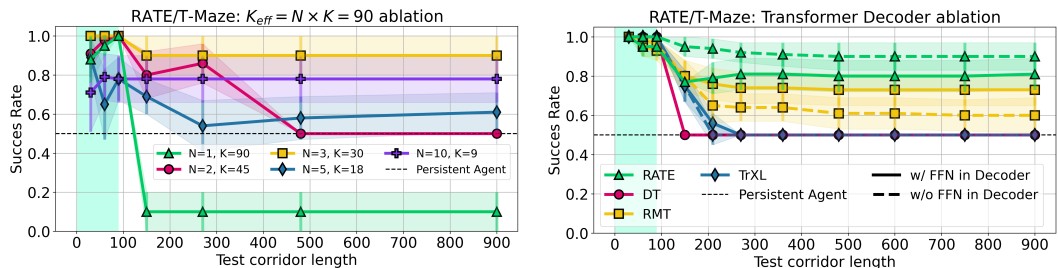

Figure 18: Ablation of segment size $K$ and segment count $N$ with fixed effective context $K_{\text{eff}} = K{\uparrow} \times N{\downarrow} = 90$.

Figure 19: Ablation of feed-forward block usage in the decoder.

3. **Transformer core.** Set the standard architecture parameters (number of layers, attention heads, embedding dimension, etc.) based on the task complexity and computational constraints.

4. **Memory tuning.** After adjust, fine-tune memory-related parameters if needed (e.g., `num_mem_tokens`, `mem_len`, dropout rates).

This configuration provides a strong default setup and has consistently performed well across all evaluated tasks.

# I  Technical details

Table 12 and Table 13 shows the technical parameters of the training models. Note that the difference between the number of DT and RATE parameters is small. Training RATE with trajectory splitting into $N$ segments allows $\sim N$ smaller GPU memory size usage than for DT. The training was conducted using a single NVIDIA A100 80 Gb graphics card.

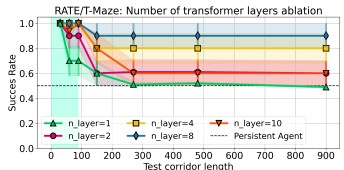
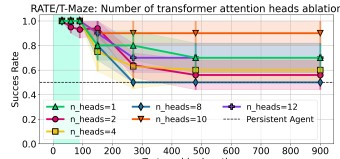
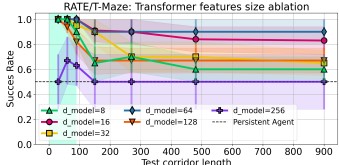

Figure 20: Results of ablation by the number of layers of the RATE model in T-Maze environment.

Figure 21: Results of ablation by the number of attention heads of the RATE model in T-Maze environment.

Figure 22: Results of ablation by the features sizes of the RATE model in T-Maze environment.

Table 12: Comparison of RATE and DT Model Parameters. RATE has 1.0-7.7% less parameters compared to DT due to the fact that RATE does not use feed-forward network in the transformer decoder by default.

| Environment | RATE | DT | diff, % |
|---|---|---|---|
| T-Maze | 1,723,840 | 1,775,488 | -2.91 |
| ViZDoom-Two-Colors | 4,537,504 | 4,672,032 | -2.88 |
| Minigrid-Memory | 2,000,864 | 2,051,872 | -2.49 |
| Memory Maze | 1,639,840 | 1,673,696 | -2.02 |
| POPGym | 6,760,192 | 6,827,008 | -0.98 |
| MIKASA-Robo | 1,412,520 | 1,529,896 | -7.67 |

Table 13: Computational efficiency comparison between RATE and DT models across different memory-intensive environments. We report three key metrics: (1) training time per epoch (mean±std, in seconds), (2) inference latency per step (mean±sem, in milliseconds), and (3) GPU memory footprint (in MiB). Lower values indicate better efficiency.

| | RATE | | | DT | | |
|---|---|---|---|---|---|---|
| Environment | Train (s) | Test (ms) | Size (MiB) | Train (s) | Test (ms) | Size (MiB) |
| T-Maze | 16.17±2.75 | 7.20±0.31 | 3,148 | 95.75±0.49 | 10.69±0.14 | 8,608 |
| ViZDoom-Two-Colors | 77.44±3.56 | 10.35±0.52 | 7,750 | 68.18±1.56 | 10.45±0.41 | 14,046 |
| Minigrid-Memory | 33.74±2.65 | 9.94±2.24 | 4,102 | 16.77±1.37 | 10.43±2.84 | 4,298 |
| Memory Maze | 110.26±2.97 | 38.98±0.62 | 6,638 | 82.69±1.56 | 40.36±0.46 | 10,386 |
| POPGym | 3.37±0.25 | 8.91±0.37 | 5,948 | 3.64±0.53 | 8.98±0.32 | 10,696 |
| MIKASA-Robo | 71.30±8.08 | 485.67±8.75 | 10,396 | 44.90±6.16 | 473.29±5.97 | 29,902 |

