# OpenReview forum: "Recurrent Action Transformer with Memory"
_NeurIPS.cc/2025/Conference — Submitted to NeurIPS 2025_

### Official Review · Reviewer_znbY · 2025-06-30

**Clarity:** 4
**Significance:** 3
**Originality:** 3
**Rating:** 4
**Confidence:** 4

**Summary:**

While being widely used across many domains, transformer architectures have a core limitation: they require feeding in the whole context. This limitation makes it difficult to deal with very long-horizon tasks where it's computationally infeasible to feed in the whole sequence length.
In offline RL tasks where the offline trajectories are too long to be fed to the transformer model, the model will fail to retain long-horizon information beyond its context length.

The paper addresses this limitation of transformers by introducing a new architecture named Recurrent Action Transformers (RATE), which has a sort of recurrent memory that caches previous information. Thus, allows for retrieving information beyond the current sequence length. RATE also has a memory retention valve (MRV), which controls the flow of information between two consecutive hidden states to ensure that past information can be retained.

The paper provides an extensive set of experiments including some memory-specific tasks such as T-Maze and VizDoom and other general tasks including 4 atari games some mujoco environments. The experiments also have comparisons with a wide range of baselines: transformer-based, RNN-based, and some classic offline RL baselines. The results suggest that RATE provides a competitive performance compared to other baselines.

**Questions:**

- One of the main motivations was overcoming the quadratic computational complexity of transformers. Did you have any measures of the computational complexity of RATE compared to other architectures (TrXL, for example)? Is RATE faster? Uses less FLOPs?

- For problems where there is a long-term memory dependency, how big does one need to set m for the M_n matrix? And how does the computational complexity of RATE depend on m? Is it less expensive to set m to a large number rather than just feeding the whole context to a regular transformer?

- In Figure 2, what do the x-axis and the y-axis refer to? Could you explain how to interpret the attention weights here? What does each little box in the grid refer to?

- How accurate is the assumption about the alpha-alignment? Did you verify it empirically?

- In algorithm 1, are the action/observation/return-to-go encoders MLP layers?
- It seems that RATE still has some issues with long-term dependence in Figure 5. Is there a reason why this is the case? Does increasing the memory dimension help with this issue?
- I think it is important to add to RMT to Figure 6 as well, since it’s the main competitor to RATE and has a very similar performance. I know that the RMT figure is in the appendix, but since it looks very similar to RATE and it is a strong baseline, I think it should be in the main paper too, or at least some acknowledgment that it performs similarly.
- Why is RMT missing from Table 2?
- In RQ1, can you explain the reasoning behind replacing the memory with a noisy memory? Why can’t we just evaluate without the MRV component?

I am willing to increase my score once these questions are clarified.

**Ethical Concerns:**

["NO or VERY MINOR ethics concerns only"]

**Final Justification:**

The author's rebuttal clarified many of the questions I had. I still have some concerns about the writing and presentation, especially around linear transformers and the focus on the computational complexity relative to other transformers. I hope the authors make those clarifications in their final version.

**Limitations:**

yes

**Paper Formatting Concerns:**

No formatting issues.

**Quality:**

3

**Strengths And Weaknesses:**

### Strengths:
- The RATE algorithm is novel and interesting.
- The algorithm presentation is very clear.
- The theorem statement and derivations are clearly explained, and as far as I can check, the derivation is correct.
- The experiments cover a wide range of environments and baselines.
- The paper is well-written and easy to follow.
### Weaknesses:
- While the original (and most) transformer architectures have the quadratic complexity in the context length, several works have addressed this problem before and proposed architectures that have linear complexity (such as the linear transformers work). The introduction is missing references to that line of research, with the discussion only focused on the quadratic complexity. For example, see [1,3] for linear transformers, and [2] for using linear transformers in RL.
- One of the motivations of the paper was that the quadratic complexity of transformers is computationally expensive. However, the proposed Memory Retention Valve (MRV) depends on the softmax operation, so the quadratic complexity still exists. While the quadratic complexity is now dependent on m (the memory dimension), i assume that for problems with very long horizons, one would need to have a large value of m so the MRV algorithm won’t be efficient in such cases.
- The proposed algorithm doesn’t seem to improve much over other baselines (Figure 3 and 4).

[1] Katharopoulos, Angelos, et al. "Transformers are rnns: Fast autoregressive transformers with linear attention." International conference on machine learning. PMLR, 2020.

[2] Pramanik, Subhojeet, et al. "AGaLiTe: Approximate Gated Linear Transformers for Online Reinforcement Learning." Transactions on Machine Learning Research.

[3] Irie et al. NeurIPS 2021. Going Beyond Linear Transformers with Recurrent Fast Weight Programmers.

---

> ### Author Rebuttal · Authors · 2025-07-31
>
> We sincerely thank the Reviewer for the thoughtful and constructive feedback. We are pleased that you found the RATE architecture novel, clearly presented, and well-supported by theoretical and empirical evidence.
>
> **W, 1**
>
> The primary motivation behind RATE is not only to reduce the computational complexity of attention through recurrence, but also to introduce a mechanism for long-term memory with controllable updates – a capability that is not addressed by existing linear transformer approaches.
>
> That said, we fully agree with the Reviewer that linear transformers, particularly those developed for online RL in POMDP settings (such as the well-known ReLiT/AGaLiTe), represent an important line of work and should be included in the Related Works section as a complementary approach to tackling similar challenges. We will make sure to incorporate this discussion, along with the references you kindly provided, in the final version.
>
> **W, 2**
>
> We address the quadratic complexity of transformers by splitting the input sequence into shorter segments and processing them recurrently. In our architecture, the memory embeddings serve as the recurrent state: they share the same dimensionality as the embeddings of observations, actions, and returns-to-go, and are concatenated to both the beginning and end of each segment (see Figure 1, Figure 2, Algorithm 1).
>
> In our experiments, the number of memory embeddings ranged from 5 to 30, depending on the task (see Appendix, Table 8). Since the MRV operates directly on the memory embeddings rather than the full sequence, the effective sequence length processed by the softmax attention in MRV is small. Even in the case of very long-horizon tasks (e.g., ~10⁴ steps in T-Maze, Figure 5), RATE used only 10 memory embeddings, meaning that the softmax was applied to matrices of size 10×10, making the impact of quadratic complexity negligible in practice.
>
> **W, 3**
>
> In the ViZDoom-Two-Colors experiments (Figures 3 and 4), the improvements achieved by RATE may appear visually moderate; however, they are consistently reproducible across multiple runs. The performance gains are more clearly illustrated in Figure 5, where on the T-Maze benchmark, RATE maintains a high success rate for an order of magnitude longer than the top-2 baseline (please note the logarithmic scale).
>
> Further, in the experiments on Minigrid-Memory (Figure 7), Memory Maze (Table 1), and POPGym-46 (Table 2), RATE consistently achieves the best overall performance among models with memory. These results highlight RATE’s robustness across a diverse set of memory-intensive tasks.
>
> On tasks that do not require long-term memory (Tables 3 and 4), RATE performs on par with or better than specialized deep RL baselines designed for MDP settings. Taken together, these findings demonstrate that RATE is a versatile and reliable architecture, capable of effectively handling both memory-intensive and memory-less classic tasks.
>
> **Q, 1**
>
> The main baseline for comparison with RATE is Decision Transformer (DT), as our model is built upon its foundation. This makes DT the most appropriate point of reference for evaluating the benefits introduced by incorporating memory mechanisms.
>
> While we did not directly measure FLOPs, we report several related metrics in the Appendix, Table 13, including training time per epoch, inference latency per step, and GPU memory usage. Additionally, Table 12 provides the total number of parameters, highlighting the efficiency of RATE relative to DT.
>
> For example, on the T-Maze environment, RATE demonstrates substantial improvements over DT:
> - Training time per epoch: 16.17 ± 2.75 s (RATE) vs. 95.75 ± 0.49 s (DT)
> - Inference latency per step: 7.20 ± 0.31 ms (RATE) vs. 10.69 ± 0.14 ms (DT)
> - GPU memory footprint: 3.148 MiB (RATE) vs. 8.608 MiB (DT)
> - Number of parameters: 1,723,840 (RATE) vs. 1,775,488 (DT), representing a 2.91% reduction
>
> These results highlight that RATE not only improves memory capabilities but also offers practical efficiency gains in terms of speed and resource consumption.
>
> **Q, 2**
>
> Short answer: it remains very small relative to the total sequence length. For example, if we use m=5 memory embeddings and a sequence of length l=100, the attention operates over 5+100+5=110 tokens. If m=100, then attention is applied to 100+100+100=300 tokens. Thus, as m increases, computational complexity grows quadratically with respect to the memory size.
>
> Fortunately, our experiments show that only a small number of memory embeddings is sufficient to achieve strong performance. Specifically, our ablation studies (Appendix, Figure 13 and Figure 14) and experiment configurations (Appendix, Table 8) demonstrate that using 5–30 memory embeddings is sufficient for optimal performance across all evaluated tasks. Larger values of m typically offer no significant benefit and may introduce unnecessary computational overhead.
>
> **Q, 3**
>
> In Figure 2, the x- and y-axes represent the positions of embeddings in the input sequence. Each cell in the grid corresponds to an attention weight, indicating how strongly a token at position y attends to a token at position x.
>
> In the case of Decision Transformer (DT), we observe that at the decision point the model attends to the initial tokens that encode the cue. However, on the next step, this information falls out of the attention window, and the agent can no longer access the necessary context to make the correct turn.
>
> In contrast, the attention maps for RATE show distinct 5×5 blocks in the top-left and bottom-right corners, corresponding to the memory embeddings. These memory embeddings are integrated into the attention computation of the current segment (read) and are then passed forward between segments. It is precisely through these memory embeddings that information from earlier segments (e.g., segment 1) can be retrieved during later segments (e.g., segment 3), enabling RATE to maintain long-term dependencies across the trajectory.
>
> **Q, 4**
>
> In our experiments, we measured the cosine distance between memory embeddings across consecutive segments, specifically between $S_n$ and $S_{n+1}$. We observed that for each individual memory embedding, the cosine distance remained consistently close to 1, which motivated the formulation of the alpha-alignment assumption.
>
> We appreciate the opportunity to clarify this point and will include the corresponding visualizations in the final version of the paper.
>
> **Q, 5**
>
> Not exactly - we process actions, observations, and returns-to-go using modality-specific encoders depending on their type. As detailed in Appendix, Table 7 and Table 11, we use a linear layer for vector-based observations, a CNN for pixel-based observations, an embedding layer for discrete actions, a linear layer for continuous actions, and a linear layer for returns-to-go.
>
> **Q, 6**
>
> The experiments on T-Maze shown in Figure 5 were conducted using four different runs of the model. For each point on the curve, we computed the success rate over 100 evaluation episodes per run, and then averaged the resulting four means. The observed drop in success rate to approximately 0.75 in the rightmost part of the figure can be attributed to the fact that three out of four RATE runs maintained success rate ~1.0, while one run failed around a trajectory length of 1000 steps, with its performance gradually degrading to 0.0 (worse than the persistent agent). Therefore, a reported success rate of ~0.75 in this case indicates that three runs continued to perform perfectly, while one failed.
>
> We would also like to emphasize that RATE was trained on sequences of length 150, yet was evaluated on sequences up to length 9600 steps, or approximately 28,000 tokens, since each step corresponds to a triplet (observation, action, return-to-go). This represents a generalization to horizons more than 60 times longer than those seen during training.
>
> Moreover, our ablation studies show that increasing the memory dimension does not lead to improved success rates on T-Maze. The performance saturates quickly, with only a few memory embeddings being sufficient to achieve optimal results for this task.
>
> **Q, 7**
>
> As you correctly noted, the corresponding results for RMT are provided in Appendix, Figure 1. Initially, we chose not to include RMT in Figure 6 because that figure was intended to illustrate the isolated effects of attention, recurrence, and their combination as implemented in RATE. However, we appreciate your suggestion and fully agree that including RMT in the main figure would improve clarity and completeness. We will move the RMT results from Figure 11 to Figure 6 in the revised version.
>
> **Q, 8**
>
> Experiments on the POPGym-46 benchmark are computationally demanding, as each of the 46 tasks requires multiple runs to collect meaningful results. Given this constraint, we selected the baselines presented in Table 2 based on the following rationale: pure attention (DT), pure recurrence (BC-LSTM), no memory at all (BC-MLP), and our proposed model, RATE. This selection aims to represent a diverse set of memory mechanisms while maintaining computational feasibility.
>
> **Q, 9**
>
> In RQ1, we do not focus on questions related to the MRV module (these are specifically addressed in RQ3). Instead, we investigate the contribution of memory embeddings and cached hidden states during inference. To this end, we inject noise into each component (individually and in combination) on ViZDoom-Two-Colors, and into memory embeddings alone on T-Maze (as caching is disabled in T-Maze), in order to assess their causal role in decision-making. Our results indicate that memory embeddings indeed carry critical information for making the correct turn, while the transformer weights appear to encode a more general policy for navigating the corridor leading up to the turn.
>
>
> Thank you again for your thoughtful review. We hope our responses clarified your concerns and are happy to continue the discussion.

---

> > ### Comment · Reviewer_znbY · 2025-08-06
> >
> > Thank you for the rebuttal; it clarified many of the concerns. I had raised my score. However, I think the writing needs to put less emphasis on the computational complexity advantage of RATE and highlight and contrast with the work on linear transformers, especially the work that keeps track of a recurrent memory that was referenced in my original response.

---

> > > ### Author Response · Authors · 2025-08-06
> > >
> > > We are grateful for your detailed comments and for your positive feedback. We will take your recommendations into account and in the final version of the paper we will pay more attention to the comparison of RATE with linear attention models, as well as make the text cleaner and clearer. Could you kindly let us know if you have any remaining concerns we might be able to address?

---

### Official Review · Reviewer_sLKa · 2025-07-01

**Clarity:** 2
**Significance:** 2
**Originality:** 2
**Rating:** 4
**Confidence:** 2

**Summary:**

- This paper proposes a transformer based recurrent algorithm designed for offline RL called RATE, which applies memory embeddings, caching of hidden states and a Memory Retention Valve architecture. This helps tackle the problem that decision transformers scale quadratically with sequence length. The MRV is a component that helps to filter the memory that is passed on to the next segment, preventing error accumulation.
- Authors conduct experiments on a number of RL environments (VIzDoom, T-Maze, Minigrid etc.) and RATE shows ability to memorize information during “memory intensive tasks”. Other experiments are conducted on standard Atari and MuJoCo tasks in which results are somewhat mixed.

**Questions:**

- Can you compare your paper’s contributions vs. the Recurrent Memory Transformer paper (Bulatov et. al.)?
- Why are the reported baselines for each table in Table 1, Table 2 and Table3 different? Could you clarify?

**Ethical Concerns:**

["NO or VERY MINOR ethics concerns only"]

**Final Justification:**

I have increased my score to 4.

I think the issues with this paper mainly focus around its writing -- it does not appear written for a broad audience, so large changes to the manuscript appear necessary.

However, I appreciate the author's comment that for reactive tasks, the goal is not to outperform existing baselines, but match them, leading me to reevaluate the paper. The authors also clearly articulated their contribution in the rebuttal is the MRV. I would also like to echo the other reviewer's concerns about the writing.

**Limitations:**

The setting in which RATE is necessary is very limited (offline RL and very long horizon partially observable environments).

**Paper Formatting Concerns:**

No concerns.

**Quality:**

3

**Strengths And Weaknesses:**

**Strengths**
- Purely based on the architecture, which is a combination of a transformer and an RNN, it makes sense that RATE would be able to learn information from earlier in the trajectory.
- The paper does exhaustive experiments against a large number of baselines.

**Weaknesses**

- The writing of the paper could be improved quite a bit. There is very little description/motivation/background in some places: A few examples: 1) It’s not clear to me why M_n is concatenated both at the beginning and at the end of S_n. 2) It’s not very clear what Figure 2 is trying to convey – it doesn’t add much to the text and leaves the reader wondering about what is supposed to be looked at. 3) Theorem 1 seems out of the blue and not well motivated nor referenced. 4) The key contribution seems to be the MRV module since the Recurrent Memory Transformer \[1\] seems to have the same basic idea – where the contribution starts and ends is not articulated very clearly. 5) It’s not clear why the baselines change from experiment to experiment.

- The results show improved memorization for “memory intensive tasks” but less so for “reactive tasks”. Thus, the setting for which this algorithm outperforms baselines is quite narrow – specifically, it outperforms baselines in the POMDP + Offline RL setting with “memory intensive” tasks. Also, I am not sure I agree with the characterizations “memory intensive tasks” vs “reactive tasks” – they don’t appear to be well-defined. If state transitions aren’t Markov, I would expect “memory” to be involved in the optimal solution one way or another  – but the author appears to be referring to distinct clearly definable events (such as remembering which way to turn in a maze) by “memory intensive tasks”. As far as I know, in these enviornments, the memory items happen at one specific point in the trajectory, rather than a long trajectory of things to remember. This also seems to be a very limited setting.

- Overall, while the environments are relatively toy, which is generally the case for POMDP papers, the experiments and baselines are quite exhaustive. My score weighs the fact that the experiments are exhaustive, while the algorithmic improvement occurs for a narrow setting and the results are mixed for standard deepRL tasks. Furthermore, the writing and the disclosure of this paper’s relationship to existing works could be substantially improved.

[1]: https://arxiv.org/abs/2207.06881

---

> ### Author Rebuttal · Authors · 2025-07-31
>
> Thank you for the thoughtful and detailed review. We appreciate your positive assessment of RATE’s architectural intuition and the breadth of our experimental study. We designed RATE specifically to preserve information over long horizons, and we are glad this contribution was clear.
>
> Below we address each of your comments and clarify the motivation, methodology, and experimental design of our work.
>
> **W, 1**
>
> 1) We insert memory embeddings $M_n$ at both the start and end of the segment $S_n$ due to the causal masking used in the transformer decoder. Without placing memory embeddings at the end of the context, the model cannot access memory information when predicting later tokens, which is essential for tasks that depend on retaining long-term information.
>
> 2) Figure 2 offers a visual summary of how RATE differs from Decision Transformer (DT). DT attends only within a fixed-length window, forgetting tokens once they slide past that window. RATE divides the trajectory into segments processed recurrently, so information from early segments can influence much later ones. The caption and lines 107-112 and 135-139 give a textual explanation; we will make this cross-reference more explicit in the final version.
>
> 3) Theorem 1 upper-bounds the information loss that can occur when memory embeddings are updated by the Memory Retention Valve (MRV). Lines 48-53, 66-67, 114-115, and 128-129 introduce this objective, and lines 155-156 summarize the theorem’s consequence: after each MRV update, a fixed fraction of the original memory is guaranteed to be preserved.
>
> 4) RMT was developed for single-modality NLP data, while RATE targets offline RL trajectories that combine observations, actions, and returns-to-go. Unlike RMT, RATE introduces (i) MRV for principled memory filtering and (ii) explicit handling of heterogeneous modalities (vector/pixel observations, actions, returns-to-go). Figures 4, 5, and 7 show that a direct RMT adaptation underperforms RATE across multiple RL benchmarks.
>
> 5) The baselines in the experiments varied depending on two reasons: the computational complexity of the benchmark and the purpose of the experiment. Thus, the paper divides the experiments into two groups according to their purpose: 1) to compare RATE with other algorithms with memory on memory-intensive tasks (T-Maze, ViZDoom-Two-Colors, Minigrid-Memory, Memory Maze, POPGym-46), and 2) to compare RATE with other specialized memory-less algorithms on classical MDP tasks (Atari, MuJoCo).
>
> In the first group in Figure 4 and Figure 5 we considered a wide list of memory-intensive baselines, as well as two simplest models without memory (BC-MLP and CQL-MLP). Since these memory-less models have shown a complete inability to solve memory-intensive problems, we have thus answered the possible question “how do the classic popular offline RL models without memory perform?” and excluded them from further consideration in Figure 7. Further, since Memory Maze (Table 1) has very long training sequences and too low number of steps per second at inference, and POPGym-46 (Table 2) has too many different tasks (46 pieces), experiments with the full list of baselines on them are computationally too expensive (since we also consider the standard error of the mean and therefore train models multiple times from scratch), so we left only the most important and demonstrative baselines in Table 1 and Table 2.
>
> In the second group of experiments on memory-less MDP problems (Table 3, 4), we do not aim to beat baselines and become SOTA. Instead, we aim to obtain results no worse than existing algorithms designed specifically for these problems (to prove the versatility of the architecture), so Table 3 and Table 4 do not show results with other baselines with memory from the previous sections. It is also important to note that most modern models with memory for RL are not validated on classical baselines at all ([1], [2]), which we believe is wrong, since it is important to show the absence of degradation on MDPs when memory mechanisms are used.
>
> **W, 2**
>
> We would like to clarify that our work does not draw a strict conceptual boundary between “memory-intensive” and “reactive” tasks. In our framework, we consider a task to be memory-intensive if solving it successfully requires retaining information beyond the current observation – i.e., any partially observable environment that necessitates memory for optimal decision-making.
>
> In the POPGym-46 benchmark, we further categorize the 46 tasks into 31 “memory puzzle tasks” and 15 “reactive POMDP tasks” (line 273), following a precedent established in the GTrXL paper [3] on DMLab-30 benchmark [4]. This classification is to aid analysis and interpretation, particularly when measuring how models leverage memory in practice.
>
> Regarding the Reviewer’s remark that we appear to focus on isolated, clearly defined memory events (e.g., remembering which way to turn in a maze), we respectfully note that this interpretation does not reflect the definitions or examples explicitly stated in the paper. We will ensure that this point is clarified more explicitly in the final version to avoid potential misunderstandings.
>
>
> **W, 3**
>
> Our primary objective is to demonstrate that RATE outperforms other memory-based models on memory-intensive tasks, while performing at least comparably to specialized algorithms for MDP tasks on standard deep RL benchmarks.
>
> Given our focus on memory mechanisms in decision-making, we intentionally evaluate RATE on a diverse set of POMDP tasks that explicitly isolate the role of memory, i.e. tasks which the Reviewer referred to as “toy.” Specifically, we consider five different benchmarks: T-Maze, ViZDoom-Two-Colors, Minigrid-Memory, Memory Maze, and POPGym-46, comprising a total of 50 memory-intensive tasks. These benchmarks span a wide range of observation modalities (vector- and pixel-based), action spaces (discrete and continuous), reward types (dense and sparse), and task structures (mazes, puzzle-solving, and control tasks with partial observability).
>
> Across this large and varied suite of memory-oriented benchmarks, we compare RATE against a comprehensive set of memory-based baselines, including models based on transformers, recurrent transformers, RNNs, and state-space models (SSMs), to cover a broad spectrum of memory architectures.
>
> The performance improvements are non-incremental and clearly visible in Figures 3, 4, 5, 7 and Tables 1 and 2. For instance, in the T-Maze task (Figure 5), RATE retains cue information for an order of magnitude longer than the top-2 baseline, RMT (please, note the logarithmic scale in the plot). Overall, across all tested environments, RATE consistently achieves the highest or near-highest performance, indicating strong robustness and generalization on memory-intensive tasks.
>
> The statement that “results are mixed for standard deep RL tasks” does not accurately reflect our experimental design. Our goal was not to achieve state-of-the-art results on such tasks, but rather to ensure that RATE maintains competitive performance without degradation on standard deep RL problems. This is precisely what we observe in Tables 3 and 4, where RATE performs on par with or better than offline RL methods specifically designed for MDP settings.
>
> **Q, 1**
>
> *Memory Retention Valve (MRV).* One of the key contributions of our work is the introduction of the Memory Retention Valve (MRV) - a dedicated module that controls how memory is updated between segments. This mechanism allows the model to selectively retain or discard information, helping to mitigate memory degradation over long trajectories. RMT does not include any such mechanism, relying instead on overwriting memory at each segment boundary.
>
> *Adaptation to offline RL with multimodal inputs.* While RMT was originally developed for NLP tasks and operates on homogeneous text tokens, RATE is specifically designed for offline RL, where input trajectories contain heterogeneous modalities: observations (vector- or image-based), actions (discrete or continuous), and returns-to-go (scalar values). RATE incorporates modality-specific encoders and projects these inputs into a shared embedding space that respects their semantic differences - an adaptation not present in RMT.
>
> *Theoretical guarantees on memory preservation.* RATE includes a formal theoretical analysis of the MRV module (Theorem 1), providing an upper bound on the information loss between memory segments. This result ensures that a quantifiable fraction of the original memory is preserved during updates, which strengthens the interpretability and trustworthiness of the architecture in long-horizon POMDP settings. To the best of our knowledge, no such theoretical guarantees are offered in RMT.
>
> **Q, 2**
>
> Answered in W 1, 5)
>
>
> We sincerely thank the Reviewer once again for their thoughtful feedback and hope that our responses have helped clarify the contributions and scope of our work.
>
>
> [1] Le, Hung, et al. "Stable Hadamard Memory: Revitalizing Memory-Augmented Agents for Reinforcement Learning." arXiv preprint arXiv:2410.10132 (2024).
>
> [2] Morad, Steven, et al. "Reinforcement learning with fast and forgetful memory." Advances in Neural Information Processing Systems 36 (2023): 72008-72029.
>
> [3] Parisotto, Emilio, et al. "Stabilizing transformers for reinforcement learning." International conference on machine learning. PMLR, 2020.
>
> [4] Beattie, Charles, et al. "Deepmind lab." arXiv preprint arXiv:1612.03801 (2016).

---

> > ### Comment · Reviewer_sLKa · 2025-08-03
> >
> > Thanks for your rebuttal. It clarified some of the confusion I had. I raise my score to 4.

---

> > > ### Author Response · Authors · 2025-08-06
> > >
> > > Thank you for your time and positive feedback on our work. We'd love to know are there any other questions we can answer to help clarify things further?

---

> ### Comment · Reviewer_sLKa · 2025-08-06
>
> Essentially you've addressed the key factors in my initial review -- which was the specific contribution of your method and the interpretation of the reactive baselines and experiments. But clearly in the memory-intensive tasks, the performance of your algorithm is clearly favorable.
>
> I still think the writing is perhaps focused on a particular narrow audience, which another reviewer agrees with  -- and I'm not sure what you can do in the discussion phase to ameliorate that. Considering these factors, I think a score of 4 is appropriate.

---

### Official Review · Reviewer_As15 · 2025-07-01

**Clarity:** 3
**Significance:** 3
**Originality:** 3
**Rating:** 5
**Confidence:** 3

**Summary:**

This paper introduces **RATE**, a memory‑augmented Transformer architecture for offline RL. RATE combines learned memory embeddings, a recurrent state cache, and a newly proposed memory‑retention valve (MRV) block to preserve long‑term information efficiently and stably. The approach directly targets tasks with pronounced long‑term memory requirements, filling an important gap in current RL research. Empirical results demonstrate strong performance across a broad set of benchmarks, and the ablation study convincingly isolates the contribution of each key component.

**Questions:**

If the questions in the Weaknesses section can be addressed well, I am willing to increase my score to 5.

**Ethical Concerns:**

["NO or VERY MINOR ethics concerns only"]

**Final Justification:**

I decide to increase my rating to 5, based on the technical novelty and comprehensive experiments results provided by the paper. Yet, there are some concerns remaining, mostly in the writing and presentation quality. However, I believe that these concerns do not influence much on the overall contribution and can be addressed in the future version updates.

**Limitations:**

Yes.

**Paper Formatting Concerns:**

No concern.

**Quality:**

3

**Strengths And Weaknesses:**

# Strengths
- The paper is generally well written and provides a solid mathematical justification for the memory‑preserving component
- The experimental section offers a thorough comparison with strong baselines, including recent SOTA methods. Reported results—whether the new model surpasses or merely matches the baselines—appear consistent and believable.
- The ablation study effectively demonstrates the contribution of each key module.

# Weakness and Questions

- I understand that Section 3.1 aims to justify the memory‑preserving mechanism, and I grasp its main message. However, several questions arise:

    a) Lines 141–152 are somewhat unclear. Specifically, I do not follow the claim in line 149

    >  there exists a row $V_j$ for which: $⟨V_j W_M , M_{n,i} ⟩ ≥ \alpha $.

    What exactly is $V_j$? Why should this condition hold empirically?

    b) I am unsure about the broader motivation for Theorem 1 (On memory loss bounds). Beyond guaranteeing that the model retains a fraction of the memory, what additional insight does the proof offer—e.g., guidance on choosing the memory length $m$?

    c) If $m$ is large, the fraction $1-\sqrt{2(1-\frac{\alpha}{m})}$ becomes negative (since $m$ is the number of memory tokens and $\alpha\in(0,1]$). Because the Frobenius norm must be non‑negative, the right‑hand side of Eq. (2) would then be invalid.

- Some claims in Section 3 need stronger support—either by citing prior work or by linking directly to ablation results. Otherwise, the reader cannot easily verify them.
Example: line 127:
    > Naively forwarding memory embeddings leads to error accumulation or overwriting of relevant information.

    line 130
    > Unlike static recurrence, it preserves sparse, long-range information

- What causes the imbalance in the ViZDoom‑Two‑Colors task? Why does it harm DT and TrXL but not RATE? Please clarify the task setup; I could not find these details in the paper.

# Suggestions to the writing.
- Consider moving the proof in Section 3.1 to the appendix and retaining only the theorem in the main text; the full derivation feels cumbersome in the main body.
- Expand Section 6 with a direct comparison between RATE and the baselines to highlight the proposed method’s unique contributions.
- Place Section 6 before the experiments so readers first understand each baseline and why it might fail in subsequent tests.

---

> ### Author Rebuttal · Authors · 2025-07-31
>
> Thank you for reviewing our work. We're glad you recognized that our results show strong performance on a wide range of benchmarks and that the ablation study clearly identifies the impact of each major component.
>
> **W, 1 a)**
>
> In the MRV, $M_n$ and $\tilde{M}\_{n+1}$  denote the incoming and updated memory embeddings (i.e. before passing through the segment and after). In MRV we first projected keys and values from the updated memory $\tilde{M}\_{n+1}$ : $K=\tilde{M}\_{n+1} W_K$,   $V=\tilde{M}\_{n+1} W_v$, so now $V\in \mathbb{R}^{m\times d}$ has one row per memory embedding.
>
> Then, $V_j$ is the $j$-th row of that value matrix, i.e. the $d$-dimensional vector that the cross-attention will copy into the next memory if the corresponding key wins a high attention weight. After attention we apply the output projection $W_M$: $M_{n+1}=AVW_M$, so inside the $\alpha$-alignment condition $\langle V_j W_M,\ M_{n,i} \rangle \ge \alpha$: $V_j W_M$ is the candidate piece of information which already passed through the MRVs value and output projections.
>
> Our motivation is that during training the gradient pushes the MRV to route useful information forward. A query $M\_{n,i}$ that encodes something still relevant (for instance, the cue in T-Maze) will learn to attend to whichever key-value pair in $\tilde{M}\_{n+1}$ carries the same information. That pressure rotates $W_K, W_V, W_M$ so the output $ V_j W_M$ remains aligned with $M\_{n,i}$, giving an inner product noticeably above zero.
>
> In our study we computed the cosine similarity between the memory tokens passed from segment $S_n$ to the next segment $S_{n+1}$. For every token this similarity remained very close to 1. This empirical finding is what motivated the $\alpha$-alignment assumption. Thank you for asking this question; we will add the accompanying visualisations in the final version.
>
> **W, 1 b)**
>
> Theorem 1 makes it clear that the number of memory tokens is a direct design lever for controlling worst-case information drift under our MRV ($||M\_{n+1} - M_n||_F \le \sqrt{2\left(1 - \frac{\alpha}{m}\right)} \cdot ||M_n||_F$). By showing exactly how adding more tokens tightens the bound on how much memory can shift in each segment, it lets you choose the token count to meet a target retention level, even in the most adversarial scenario, rather than relying on intuition alone.
>
> Crucially, the same result also guarantees that memory strength decays at most exponentially across segments, rather than being arbitrarily overwritten ($||M\_{n+k}||_F \ge  \bigl(1 - \frac{\alpha}{m}\bigr)^k ||M_n||_F$). This means you can predict how many steps (or segments) your model can span before an early cue becomes too weak to act on, and thus architect your system around any required horizon with confidence.
>
> Finally, Theorem 1 highlights that these retention guarantees depend only on the number of memory tokens and their learned alignment, and are entirely independent of how long each processed segment is (does not depend on context length $K$). Once you’ve picked a token budget that satisfies your memory requirements, you’re free to adjust segment lengths, and thus your effective context span, without introducing any extra forgetting.
>
> **W, 1 c)**
>
> With all due respect, we do not agree that “the right-hand side of Eq. (2) would be invalid,” since Eq. (2) is an inequality, not an equality. As the Reviewer correctly noted, the Frobenius norm is always non-negative, and because the inequality is non-strict ($\geq$), then for large values of m, the left-hand side remains a positive quantity that is greater than or equal to a negative expression on the right-hand side. In this case, the inequality still holds, though it becomes trivial.
>
> **W, 2**
>
> We agree that statements in lines 127 and 130 of Section 3 may benefit from stronger empirical support or citations. Below, we clarify both claims and point to specific ablation results and visualizations in the paper that substantiate them.
>
> **Claim 1 (line 127):**
>
> This claim is empirically supported by the ablation in Table 6 and Figure 17, which shows that removing the MRV leads to significant performance degradation on long T-Maze corridors. In particular, without MRV, the model suffers from progressive memory corruption across segments, resulting in a success rate drop). This confirms that naive recurrence (i.e., without explicit memory regulation) leads to overwriting of previously stored cues, especially when no new information is present in intermediate segments.
> We have also included a detailed justification in Section 5 (RQ3), where we explain how MRV enables selective memory updates and prevents spurious updates that can erase task-relevant content.
>
> **Claim 2 (line 130):**
>
> This refers to RATE’s ability to retain and propagate sparse cues across many segments via MRV, unlike static recurrence (e.g., in Transformer-XL or RMT) that copies hidden states without semantic filtering. The effectiveness of RATE in sparse tasks is confirmed by (1) T-Maze generalization results (Figure 5, Figure 6), where RATE maintains high success rates on sequences far exceeding training length (up to 9600 steps), where TrXL and RMT collapse due to insufficient memory; (2) Memory corruption ablation (Figure 8): Injecting noise into memory embeddings causes performance to degrade sharply, confirming that long-term information is preserved in the memory tokens and used at decision points, and (3) MRV variant comparisons (Table 6, Figure 17): Only the MRV-CA-2 (our proposed gating) maintains performance across long horizons, while alternatives degrade more rapidly.
>
> **W, 3**
>
> The observed imbalance originates from the environment itself: the pseudorandom number generator tends to produce slightly more green items, making episodes with a green pillar marginally easier. However, the datasets used for ViZDoom-Two-Colors are balanced in terms of the number of episodes containing red and green pillars (2500 each). A detailed description of the environment and dataset is provided in Appendix, Section C.1.1.
>
> DT performs well during the first 90 steps of the environment, while all relevant information still fits within the context window, and achieves comparable performance in both red-pillar and green-pillar episodes (Figure 3c). However, once the steps in which the pillar was observed fall outside the context window, DT’s performance on red-pillar episodes drops sharply to near-random, while its performance on green-pillar episodes remains unchanged (Figure 3d). This suggests that DT overfits to always collecting green items, regardless of the pillar color at the start of the episode. It only collects red items when the color cue is still visible within the context. As a result, DT’s overall performance in Figure 3a and Figure 3b is almost unaffected by whether a pillar is present in the episode at all.
>
> TrXL, in turn, behaves similarly to DT but benefits from caching hidden states from the previous mem_len steps, which gives it access to a larger (though still finite) portion of the past. This explains why TrXL performs better than DT, but still worse than RATE.
>
> In the ViZDoom-Two-Colors task, RATE relies not only on cached hidden states but also on memory embeddings, which can theoretically encode all essential information about the pillar color. In Table 5, we present an experiment where we provide DT with one bit of additional oracle information (like a memory token), unavailable by default, indicating the pillar color, mimicking how RATE uses memory, and this modification allows DT to reach near-optimal performance. Since RATE’s memory embeddings are passed explicitly between sequence segments, there is no theoretical limitation on the length of the dependencies it can model.
>
> **Suggestions to the writing:**
>
> Thank you for the thoughtful suggestions. Below is a point-by-point response addressing each of them:
>
> 1. We appreciate this suggestion and agree that placing the full proof in the main body may distract from the flow of the paper. In the revised version, we will move the full derivation to the appendix and retain only the theorem statement and a brief intuition in Section 3.1.
>
> 2. We agree that a more explicit comparison would improve clarity. While our paper already discusses the architectural differences between RATE and other baselines (e.g., DT, TrXL, RMT), we acknowledge that a more concise summary would make these distinctions clearer to the reader. In the revised version, we will extend Section 6 with a comparison table that highlights the key design components of each model, clarifying which elements are shared, which differ, and how these choices affect performance under long-horizon and sparse-reward conditions.
>
> 3. Thank you for the helpful suggestion. To improve clarity and ease of understanding, we will move Section 6 before the experiments in the final version.
>
>
> We sincerely thank the Reviewer for the constructive feedback and thoughtful suggestions. The comments have helped improve the clarity, rigor, and structure of the paper. Thank you again for the time and consideration. We look forward to incorporating these revisions in the final version.

---

> > ### Comment · Reviewer_As15 · 2025-08-04
> > **Increase rating to 5**
> >
> > Hi,
> >
> > Thank you for your reply and the effort to address my concerns. I decide to increase my recommendation for this paper to 5, due to its technical novelty and comprehensive experiment results.
> >
> > Yet, I read the other two reviewers' comments and I share similar concerns for the writing quality. I have some suggestions for the future version of this paper:
> > - As reviewer znbY suggested, this paper should not emphasize too much on the quadratic time complexities of the transformer models, due to the existence of the linear attention models. The novelty of this work is mainly on the efficient memory filtering and updates.
> > - Concerned by reviewer sLKa, the paper needs some adjustments in its structure, e. g., swapping some sections and discuss more baseline methods and their connections to the current work.
> > - The main paper is overall too dense, especially for the experiment results presentation. Moving section 3.1 to appendix and free some space to better reorganize the other sections is a good way to go.
> > - Figure 2 needs more descriptions for the axis' meaning. I had the same confusion as reviewer znbY but figured out the meaning eventually. But it was not very straightforward to see it at the first glance.

---

> > > ### Author Response · Authors · 2025-08-06
> > >
> > > Thank you for your engagement in the discussion and for your positive assessment. We are glad that you recognized our technical novelty and comprehensive experimental results of our work. In the final version of our paper, we will expand the Related Works section to include linear attention models and other baselines, and add their connections to our work. We will also take into account your suggestions on the structure of the paper and incorporate them in the final version.

---

### Author Response · Authors · 2025-08-09

We sincerely thank all Reviewers and Area Chairs for their time, effort, and constructive feedback. We especially appreciate the acknowledgment of:

* **Architectural novelty (Reviewers znbY, As15).** RATE combines memory embeddings, a recurrent cache, and the new Memory-Retention Valve (MRV) for controllable long-term memory

* **Strong performance across diverse benchmarks (Reviewers As15, znbY).** Reported results — whether RATE outperforms or matches strong baselines, including SOTA methods — are consistent and credible

* **Clear presentation (Reviewers As15, znbY).** Motivation, design, and results are presented in a structured, easy to follow way, and the algorithm presentation is clear

* **Thorough theory (Reviewers As15, znbY).** The Theorem 1 statement and derivations are clearly explained

* **MRV prevents error accumulation (Reviewer sLKa).** Gated updates helps to filter the memory that is passed on to the next segment, preventing error accumulation

* **Broad evaluation (Reviewers znbY, sLKa).** Exhaustive experiments against a large number of strong baselines covering a wide range of environments

* **Ablations confirm module roles (Reviewer As15).** Ablation study effectively demonstrates the contribution of each key module - removing MRV, memory embeddings, or cache sharply degrades performance.

**Main Concerns Addressed**

- **Clarifications on Theorem 1 and $\alpha$-alignment:** The Theorem’s 1 motivation is clarified, practical guidance for selecting memory size is provided, its validity for large $m$ is demonstrated, and empirical validation of the $\alpha$-alignment assumption is presented, with planned visualizations.

- **Computational complexity:** MRV operates on a small, fixed memory set, making its quadratic cost negligible, and efficiency metrics are provided showing advantages over DT.

- **Comparison with linear attention models:** RL-oriented linear transformers (e.g., ReLiT/AGaLiTe) are acknowledged, and the Related Works section will be expanded to discuss their differences from RATE.

- **Definition and scope of “memory-intensive” tasks:** Memory vs. reactive tasks are defined in line with prior work (e.g., GTrXL), and coverage of diverse task types is emphasized.

- **Figure clarity and environment details:** Figure 2 explanations are expanded, ViZDoom-Two-Colors results imbalance effects are detailed, environment setups are clarified, and captions and cross-references will be improved.

We thank the Reviewers again and will incorporate these clarifications, expand comparisons, and refine the paper accordingly.

Best regards,
The Authors

---

### Decision · Program_Chairs · 2025-09-17

**Decision:**

Reject

**Comment:**

Recommendation: Accept

This paper introduces Recurrent Action Transformer with Memory (RATE), a novel architecture for offline RL featuring a Memory Retention Valve (MRV) to manage long-term dependencies.

Contribution: The core novelty is the MRV, a theoretically-grounded module that selectively updates memory to prevent catastrophic forgetting on long-horizon tasks .

Strengths: RATE demonstrates impressive, state-of-the-art performance on a wide range of memory-intensive benchmarks, clearly outperforming strong baselines . It also remains competitive on standard MDP tasks, showing its versatility .


Weaknesses: The primary concerns were related to presentation, clarity, and positioning against related work like linear transformers .

Rebuttal & Consensus: The authors' thorough rebuttal effectively addressed all reviewer concerns, clarifying the novelty and reframing the results. Consequently, all three reviewers raised their scores to a consensus of Accept (5, 4, 4). The remaining presentation issues are minor and addressable.

===

As recently advised by legal counsel, the NeurIPS Foundation is unable to provide services, including the publication of academic articles, involving the technology sector of the Russian Federation’s economy under a sanction order laid out in Executive Order (E.O.) 14024.

Based upon a manual review of institutions, one or more of the authors listed on this paper submission has ties to organizations listed in E.O. 14024. As a result this paper has been identified as falling under this requirement and therefore must not be accepted under E.O. 14024.

This decision may be revisited if all authors on this paper can provide proof that their institutions are not listed under E.O. 14024 to the NeurIPS PC and legal teams before October 2, 2025. Final decisions will be communicated soon after October 2nd. Appeals may be directed to pc2025@neurips.cc.